# Coastal Dynamics at Kharasavey Key Site, Kara Sea, Based on Remote Sensing Data

Georgii Kazhukalo [1,*], Anna Novikova [1], Natalya Shabanova [1], Mikhail Drugov [1], Stanislav Myslenkov [2,3], Pavel Shabanov [1,3], Nataliya Belova [1] and Stanislav Ogorodov [1]

1 Laboratory of Geoecology of the North, Faculty of Geography, Lomonosov Moscow State University, 1 Leninskie Gory, 119991 Moscow, Russia
2 Department of Oceanology, Faculty of Geography, Lomonosov Moscow State University, 1 Leninskie Gory, 119991 Moscow, Russia
3 Shirshov Institute of Oceanology, Russian Academy of Sciences, 36, Nakhimovskii pr., 117997 Moscow, Russia
* Correspondence: kazhukalo@geogr.msu.ru; Tel.: +7-495-939-2526

**Abstract:** In recent decades, acceleration of coastal erosion has been observed at many key sites of the Arctic region. Coastal dynamics of both erosional and accretional stretches at Kharasavey, Kara Sea, was studied using multi-temporal remote sensing data covering the period from 1964 to 2022. Cross-proxy analyses of the interplay between coastal dynamics and regional (wave and thermal action) and local (geomorphic and lithological features; technogenic impact) drivers were supported by cluster analysis and wind–wave modelling via the Popov–Sovershaev method and WaveWatch III. Ice-rich permafrost bluffs and accretional sandy beaches exhibited a tendency towards persistent erosion (−1.03 m/yr and −0.42 m/yr, respectively). Shoreline progradation occurred locally near Cape Burunniy (6% of the accretional stretch) and may be due to sediment flux reversals responding to sea-ice decline. Although the mean rates of erosion were decreasing at a decadal scale, cluster analysis captured a slight increase in the retreat for 71% of the erosional stretch, which is apparently related to the forcing of wind–wave and thermal energy. Erosional hotspots (up to −7.9 m/yr) occurred mainly in the alignment of Cape Kharasavey and were predominantly caused by direct human impact. The presented study highlights the non-linear interaction of the Arctic coastal change and environmental drivers that require further upscaling of the applied models and remote sensing data.

**Keywords:** coastal dynamics; permafrost; multi-temporal images; bluffs erosion; shoreline accretion; climate change; WaveWatch III; cluster analysis; Arctic; Kara Sea; Yamal Peninsula



## 1. Introduction

The complexity of Arctic coastal processes in comparison to temperate ones is due to the onshore permafrost features and spatiotemporal variability of sea-ice extent [1–4]. Despite the negligible mechanical wave action exerted by the presence of sea ice for a considerable part of the year, the average decadal-scale shoreline change rate across the entire Arctic Basin is −0.5 m/yr [3]. Shoreline change rates vary significantly due to the regional and local geomorphic, cryolithological and climatic features [1,2,5]. The peak rates of coastal erosion were observed on the Laptev (−2.2 m/yr long-term mean in 1965–2011 for three key sites [6]) and Beaufort seas (up to −13.6 m/yr in 2002–2007 [7]). Advances in remote sensing techniques provided new insights into coastal erosion approaches by creating the opportunity to investigate new regions and expand the time scale [8]. Yet, local uncertainties of climate and geomorphic drivers lead to difficulties in upscaling local-scale coastal estimations and creating global forecasting models.

The majority of the Arctic coastal erosion studies revealed an acceleration of the retreat rates since the beginning of the 21st century, at both Russian key sites [4,6,9–11] and globally [1–3,5,12]. According to climate change projections, Arctic coastal erosion rates may face up to a 2- or 3-fold increase by the end of the 21st century [13]. For example,

analysis of a 60 km stretch of the Alaskan Beaufort Sea coasts using a time series of aerial photography revealed that mean annual erosion rates increased from 6.8 m/yr (1955 to 1979), to 8.7 m/yr (1979 to 2002) and 13.6 m/yr (2002 to 2007) [7]. Hence, understanding coastal dynamics and the interplay of technogenic and environmental drivers is vital to shoreline management and climate change adaptation [14] in order to reduce subsequent damage to engineering infrastructure. This is particularly important for the study area, which is considered one of the largest gas fields in the world.

The remoteness and inaccessibility of the Kara Sea region lead to an insufficient number of study areas with multi-decadal data regarding coastal dynamics. The most drastic erodibility was observed at the Marre-Sale key site (150 km southward to Kharasavey), with an average retreat rate of 1.7 m/yr spanning from 1947 to 2001 [10,15,16]. The Ural coasts of the Baydaratskaya Bay also eroded rapidly at an average rate of 1.2 m/yr between 1964 and 2016 [11]. Kharasavey area, located on the west coast of the Yamal Peninsula, is one of the most long-term coastal monitoring fields in the Russian Arctic. Observations of coastal dynamics have been carried out by the laboratory of Geoecology of the North since 1981 [17–20]. Studies were also conducted by F.A. Kaplyanskaya [21], N.F. Grigoriev [22], V.I. Astakhov [23], I.V. Yuryev [24], Yu.K. Vasilchuk [25], A.A. Vasilyev [15,16] and many others. Based on field monitoring data, the average shoreline change (1981–2012) is approximately 1.0 m/yr [19]. According to remote sensing techniques [15,16], the average coastal retreat rate for an erosional stretch was 1.4 m/yr, and the peak rate was approximately 3.0 m/yr. Authors declare that the calculated retreat rate at Kharasavey is one of the greatest in the western part of the Russian Arctic [16]. According to other research at Kharasavey site [25], the shoreline retreat rate was estimated as 1.13 m/yr, with a maximum value of 2.3 m/yr. A recent, most detailed investigation [19,20] was applied within a southern, 9 km long area. Two coastal stretches with multi-decadal average retreat rates of 2–3 m/yr were identified. Based on these data, extreme shoreline change is linked to a high ground ice content in coastal bluffs on these segments.

Thus, much of the preceding studies were focused on shoreline change in ice-rich permafrost bluffs, the dramatic erodibility of which is caused by thermal abrasion and thermal denudation [26,27]. Only a few studies [22,28] were dedicated to accretional wave-dominated coasts, located northward to Kharasavey settlement. Such predominance of permafrost bluff investigation and the negligible degree of understanding of accretional stretches are evident throughout the entire Arctic [12,29]. Exclusive assessment of erosional coasts can skew the focus towards local drivers and may lead to the loss of regional-scale patterns of coastal behavior among coherent lithodynamic systems.

Hence, the purpose of this research is to investigate coastal dynamics along geomorphologically distinct stretches at Kharasavey key site by expanding spatial and temporal scale via high-resolution remote sensing data. The present paper provides new insights regarding coastal dynamics for both erosional and accretional coastal stretches. Using multi-temporal aerial and satellite imageries, multi- and intra-decadal shoreline change rates were observed. Results of the investigation are also based on cross-proxy analysis of remote sensing and field data supported by cluster-based segmentation and wind–wave modelling (via the WaveWatch III [30] and the Popov–Sovershaev method [31,32], which would rather improve understanding of the interplay of technogenic impacts, coastal geomorphic features and current climate trend.

## 2. The Study Area

The presented study is dedicated to an analysis of the Kharasavey coasts of Western Yamal, Kara Sea. Studied area extends 21 km of slightly concave coastline between Cape Burunniy in the north to Cape Kharasavey in the south (Figure 1). Hinterland of the key site is an alluvial-marine plain that was formed as a result of marine, lacustrine and fluvial sedimentation in the conditions of the sea-level fluctuations during the last ~50 ka in Late Pleistocene–Holocene [33]. The terrain comprises of at least three distinct levels: Holocene laidas (height is up to 2–4 m above sea level), a low alluvial-marine terrace (7–16.5 m) and a

high marine terrace (up to 30 m). Despite quite homogeneous genesis, geomorphic setting of coastal stretches are significantly distinct.

The central and northeastern parts of the hinterland are composed of a high terrace that is greatly penetrated by thermal erosion gullies and ravines, solifluction lobes, small pingos and, less commonly, permafrost thaw lakes and baydzharakhs. The highest relict of this surface occurs in the alignment of Cape Burunniy with seaward, gently sloping (6–12°) paleo bluff. The bluff toe is adjacent to wide (up to 2–3 km) Holocene laida, which forms by tides and storm surges and is widespread among the Kara Sea coastline [34,35]. The surface of laida is separated on two distinct levels based on their morphology, height and type of vegetation. High laida (height is 3–4 m) is rarely affected by storm surges and infilled with relict lagoon and thaw lakes. Further seaward, low laida (1–3 m) has sparse vegetation and is infilled with salt ponds, washover and abandoned channels. The low alluvial-marine terrace composes the southern part of the key area. The surface of the terrace is slightly eroded by gullies and mostly infilled with ice wedge polygons, thaw lakes and pingos. Ice-rich coastal bluffs of the study area are confined to that geomorphic level.

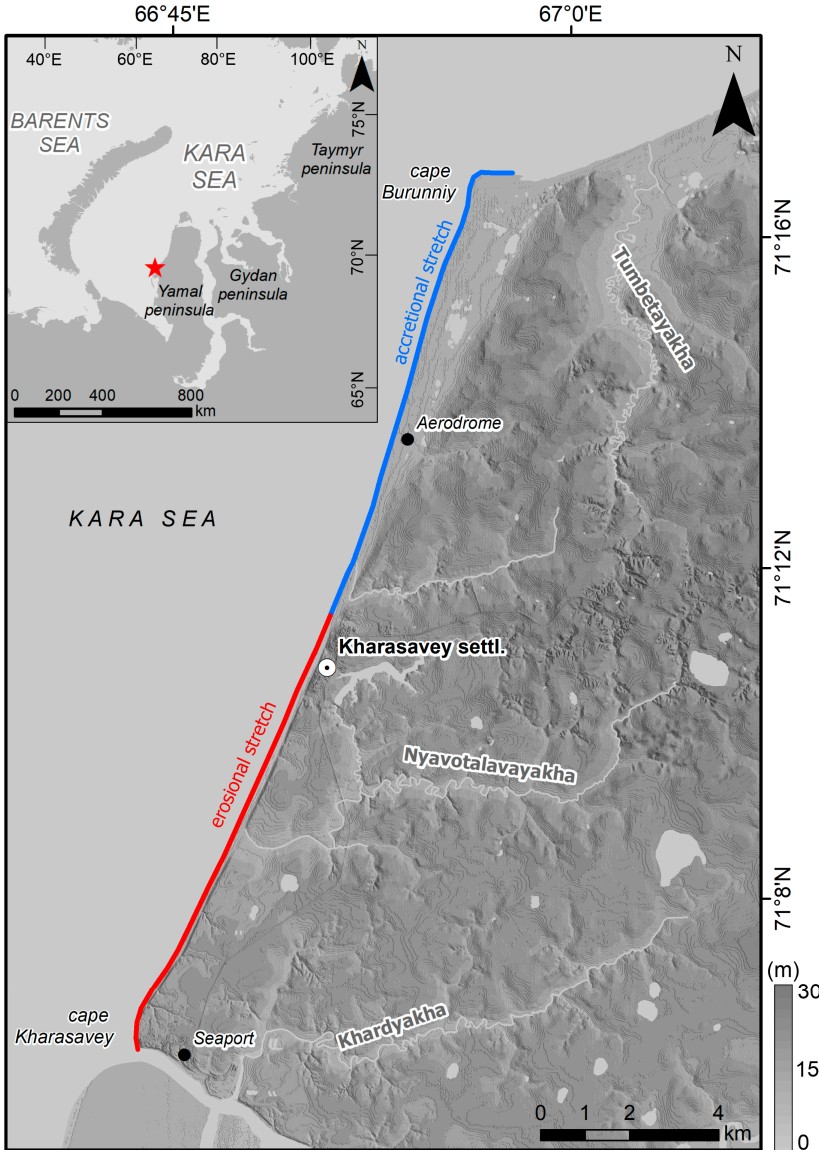

**Figure 1.** Location and topography of the study area. Red star indicates the position of Kharasavey key site among the Kara Sea. Background is ArcticDEM [36].

Coasts among the whole area are micro-tidal and wave dominated, but their morphology differs significantly. The coastline is divided into two reaches—low-lying accretional sandy beaches in the north (Figure 2a) and erosional with ice-rich coastal bluffs in the south (Figure 2b). Bluffs of the erosional stretch are undercut by thermal abrasion and thermal denudation [26,27]. The bluff steep (35–50° and above) face is highly penetrated by baydzharakhs, mud lobes, debris, retrogressive thaw thumps and gullies on ice wedge polygons (Figure 2d). The beach on these reaches is narrow (from 3–5 m up to 40–50 m near river mouths) and steep. In contrast, the beach on the accretional stretch is wide (up to 400–500 m) and gently sloping. The backshore of the accretional coasts (Figure 2c) is infilled with hummocky foredunes (height is up to 80 cm) and adjusts to the low laida surface that was outlined above. The tidal flat is usually 20–50 m wide with the exception of Cape Burunniy area (up to 200–300 m).

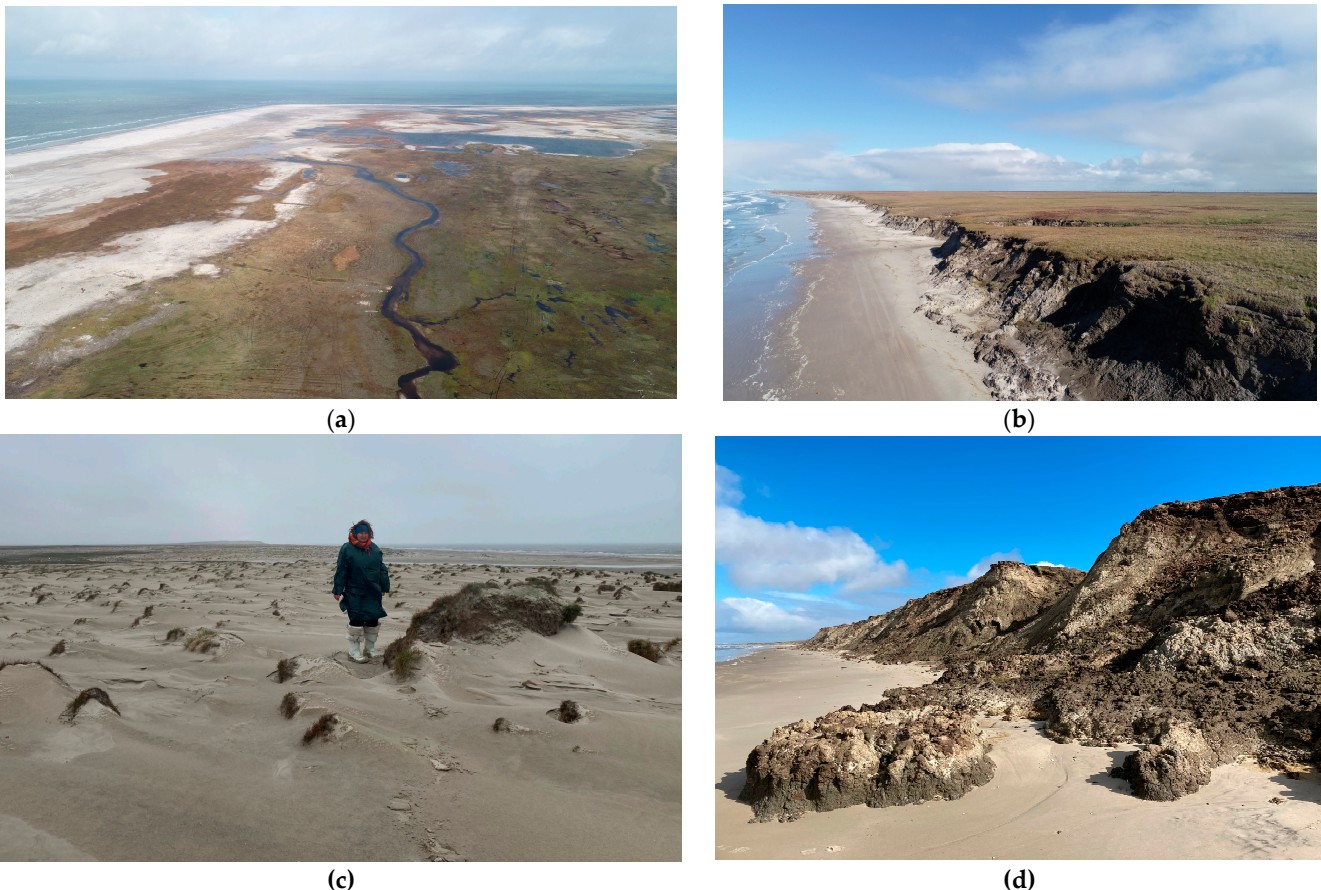

(a)     (b)

(c)     (d)

**Figure 2.** Coastal geomorphic features of the study area: (**a**) sand beach and surface of laida at Cape Burunniy; (**b**) ice-rich erosional bluffs, penetrated by thermal denudation, to the south of the Nyavotalovayakha River; (**c**) hummocky foredunes adjacent to the backshore of accretional coasts. Person for scale: (**d**) typical onshore morphology of the erosional stretches. The backshore and bluff face are infilled with mud lobes, ravines and block failure.

Offshore bathymetry is gently sloping—3 m isobath located in 3 km seaward, while 10 m isobath is traced at 8–10 km away from shoreline. The upper shoreface area is composed of series of longshore bars with a height up to 30–40 cm. Intensive beach accretion near Cape Burunniy is due to specific bathymetry. The shallowness of the offshore is caused by the wide (up to 3–3.5 km) underwater spit that is oriented to the west and presumably lies on the relict subaqueous permafrost.

The study area is entirely covered by thick Quaternary deposits with the sole at a depth of more than −150 m [37]. Holocene laidas consist of silty sand sediments with

abundance of organic material (peat). Coastal bluffs are predominantly composed of loams, silts, and sands (Figure 3), which are chaotically intertwined alongshore [19,25]. Silty and sandy marine sediments of the southern part of the key site are often compound with inclusions of lacustrine-alluvial fine sands and sandy loams, peat or alluvial detritus [19]. Offshore sediments are composed of fine and very fine silty sands [22].

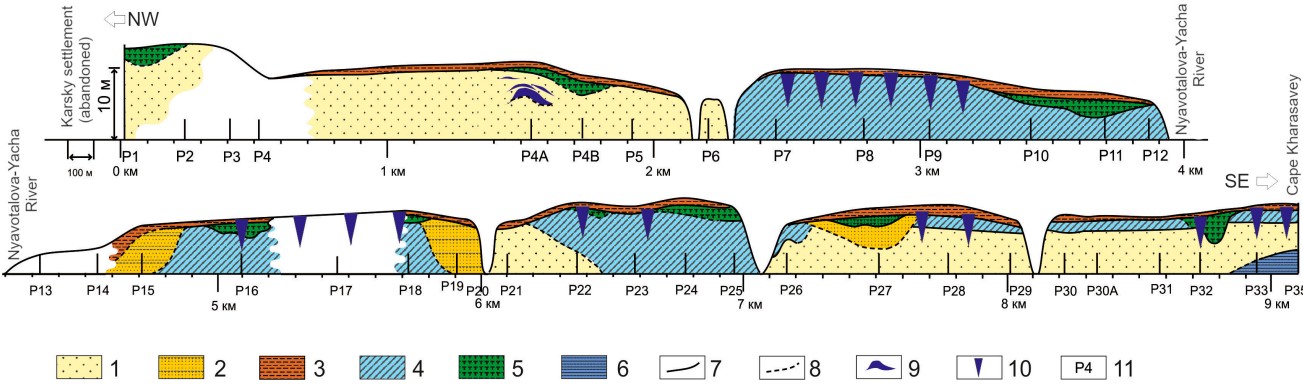

**Figure 3.** Sediment composition of the coastal bluffs between Kharasavey settlement and Cape Kharasavey [19]: 1—sands, fine and meduim-grained, with interlayers and lenses of loams, marine; 2—sands, fine-grained, with loam and peat interlayers, lacustrine-alluvial; 3—loams, sand-loams; 4—loams, marine (to the south of profile P26—of unspecified origin); 5—peat, locally with loam and sandy interlayers; 6—clays; contacts between lithologies: 7—established, 8—assumed; 9—tabular ground ice; 10—ice wedges; 11—profile number of network monitoring.

The territory is located in the continuous permafrost zone. The thickness of permafrost varies from a few m nearshore to 150–180 m on the third marine terrace. Among coastal zones, the mean annual temperature range from −2.5 °C to −8.0 °C. The thickness of the active layer is due to the composition of the sediments, surface drainage conditions and type of vegetation. Field works revealed a permafrost table at the depths of 0.2–0.3 m on laidas and 1.1–1.4 m on coastal bluffs. Permafrost conditions lead to the presence of ground ice within bluff sediments. Marine loams are greatly penetrated by epigenetic ice wedges, while sandy sediments have lower ice content (<30% of total mass) [19]. Massive ice beds rarely occur and outcrop only in the northern part of the erosional stretch [19].

Kharasavey key site is situated within a polar marine climate with annual mean air temperature of −9.7 °C (according to nearest weather station, Marre-Sale [38]. Mean July temperatures tend to increase from 7–8 °C to 10 °C since 1978 [33,38]. Predominant summer wind directions are northern, western and northwestern. Wind rates of more than 10 m/s are relatively rare, occurring up to 8% per year, and only a 15.6% of them take place during the ice-free period [39]. The average wind speed during the summer period ranges from 5.9 to 6.4 m/s. Average wave height during the ice-free period is less than 90 cm [22]. The maximum wave height with a 3% probability of occurrence in 10 years is 5 m [39], which makes it possible to cover the accretional stretches, including sandy beaches and the laida surface near Cape Burunniy. In July to August, daily tidal fluctuations are up to 55–65 cm [22,34]. Landfast ice usually occurs in October and collapse in late June [22,25]. The duration of the ice-free period in the studied region, as well as in the whole Arctic, has shown a significant increase (up to 15–75%) [40–42] during the last 40 years.

Exploitation of the study area has begun in 1976 with the construction of Kharasavey and Karskoe (abandoned) rotation camps. Since the 1970s landscapes near Kharasavey settlement were significantly transformed [43]. Engineering constructions within main settlement include sand mounds (height is up to 3 m) for residential buildings and network of drilling platforms, sand embankments (up to 5 m) for linear constructions (pipelines, roads, etc.) and the earth dam. Territory near Cape Kharasavey has also been built up with sea pier and other constructions.

Aforementioned development results in substantial technogenic transformation of the coastal and hinterland terrain. Reduced vegetation, hydraulic works and landform alteration lead to widespread intensification of surface processes, such as thermokarst, landslides, gully erosion and solifluction. Rapid gully formation within coastal zone, for example, demanded to use construction waste as a backshore sea defense [43]. The surface of laida near Cape Burunniy is penetrated by quarries and dumps of hydraulic sand fill. Previous study [20] also revealed significant excavation of beach and bluff sediments in the alignment of Kharasavey Cape Moreover, dredging works were carried out in order to increase depths near the pier. Such technogenic impact on offshore and onshore may lead to considerable changes in coastal behavior.

## 3. Materials and Methods

### 3.1. Remote Sensing Data

The presented approach is chiefly based on the analysis of multi-temporal high-resolution remote sensing data. Coastal change rates along 21 km of shoreline were observed and compared with an assessment of geological factors and hydrometeorological features (Figure 4).

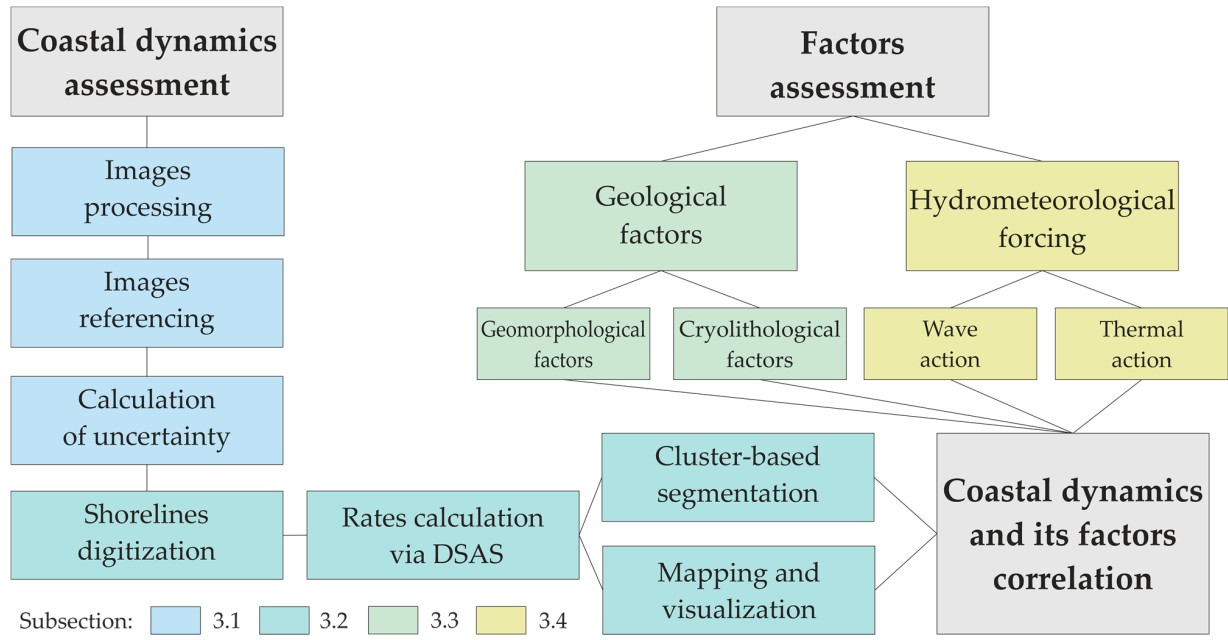

**Figure 4.** The study workflow.

Shorelines were digitized using 10 satellite and aerial imageries within 58 years (Table 1): three aerial film-typed images of the 1970–1980s, two satellite film-typed images of the 1960s and of the 1980s and five satellite digital images of the 2000–2020s. The aerial images of 31.07.1972 (1.0 m spatial resolution after scanning and referencing), 21.07.1977 (1.5 m) and 05.08.1988 (0.5 m) are soviet aerial images, obtained from the archives of the Laboratory of Geoecology of the North and Faculty of Geography of Moscow State University and scanned for further processing in GIS. Images KH-4A Corona of 08.08.1964 (5.0 m) and KH9-17 Hexagon of 26.07.1982 (1.2 m) are declassified military images of USA, provided by the U.S. Geological survey [44] as scanned film strips ALOS PRIZM of 16.07.2006 (spatial resolution is 2.0 m), GeoEye-1 of 18.07.2014 (0.4 m), WorldView-2 of 15.06.2016 (0.5 m), PlanetScope of 20.08.2020 (0.5 m) and Jilin-1 Gaofen-03 of 20.08.2022 (0.75 m) satellite images of various ownership (Maxar tech., CIOMP, JAXA) were purchased from different distributors or provided by the Geoportal MSU in digital format.

**Table 1.** Summary of the remote sensing data used in the approach.

| Sensor | Acquisition Date | Spatial Resolution (m) | Referencing Uncertainty (m) |
|---|---|---|---|
| KH-4A Corona | 08.08.1964 | 5.0 | 3.6 |
| Aerial | 31.07.1972 | 1.0 | 2.5 |
| Aerial | 21.07.1977 | 1.5 | 2.9 |
| KH9-17 Hexagon | 26.07.1982 | 1.2 | 2.7 |
| Aerial | 05.08.1988 | 0.5 | 2.4 |
| ALOS PRIZM | 16.07.2006 | 2.0 | 2.5 |
| GeoEye-1 | 18.07.2014 | 0.4 | - |
| WorldView-2 | 15.06.2016 | 0.5 | 1.7 |
| PlanetScope | 20.08.2020 | 0.5 | 1.4 |
| Jilin-1 | 20.08.2022 | 0.75 | 1.2 |

The images were processed (modern images were also pan-sharpened) and referenced all to the GE-1 image (as the most detailed and the sharpest) with Spline function with the use of ArcGIS software [45]. The uncertainty of referencing was assessed manually by measuring the distances between the test point's locations on the different images and calculating root mean square error (RMSE) for every set of points.

*3.2. Shoreline Change Determination and Analysis*

Quantifying of coastal change rates precludes determination of shoreline position on each time span. Definition of coastal dynamics indicator may be performed in various ways [46–50]. The bluff toe (bottom) and bluff top, which are evident topographic breaks, are commonly used as a shoreline proxy on erosional coasts [6,9,12,51]. To estimate coastal retreat of the accretional coasts many authors map erosional scarps if they are visible. In cases of erosional scarp absence, shoreline digitizing on low-lying coast without any other topography breaks may be challenging [47]. On the accretional stretch of the study area, four shoreline indicators may be identified in case of absence of the erosional scarps (Figure 5). Vegetation line (V) that separates sandy beach and laida surfaces is clearly detected by spectral differences, but greatly exerted by foredune migration and technogenic intervention (by extraction of sands or building construction). Expansion of the hummocky dune field (D) can indicate coastline stabilization with no significant wave impact, but is spatially and temporally sparse. Instantaneous water line (W) is strongly affected by wind/wave conditions and tidal stage at a certain time. Wet/dry line (H, matches with high tide in presented case) is one of the most common shoreline proxy [47,48]. It can be digitized based on spectral differences and is considered relatively static, less reliant on wind/wave conditions [47]. Hence, due to significant human impact on the position of backshore (V, D) proxies, shoreline for the accretional stretch is defined as wet/dry line (H). Shoreline proxy of the accretional coast on the aerial imagery of 1972 was not digitized due to the flare covering the northeastern part of the area. Proxies of 2014 and 2016 imageries have also not been traced in order to avoid short time periods (less than 6 years) due to relatively high uncertainty of accretional shoreline with no topographic break. Consequently, in the presented study, erosional scarp (bluff top) and wet/dry line are defined as shoreline proxy for the erosional and the accretional stretches, respectively.

Shoreline change analysis was performed via widely used in coastal studies [2,12,29]. Digital Shoreline Analysis System (DSAS v. 5.1 [52]) in ArcGIS environment. The program automatically builds transects at optional intervals (50 m was assigned in the presented case). Then, the set of transects was adjusted manually (on erosional stretch) to avoid river mouths. The final dataset incorporates 194 and 214 transects on erosional and accretional stretch, respectively. Rates of shoreline change in each transect were calculated as the distance between two shoreline proxies divided by the relevant time spans—so-called end point rate (EPR, m/yr). Obtained statistics of shoreline retreat and advance for different periods, mainly time-averaged EPRs, were implemented in computer code for further examination. Other statistics, such as Linear Regression Rate (LRR) or Weighted Linear

Regression (WLR), were not obtained due to the fact that the forecasting of shoreline changes was beyond the scope of this study. For the purpose of hindcast analysis, EPRs calculation is sufficient [52].

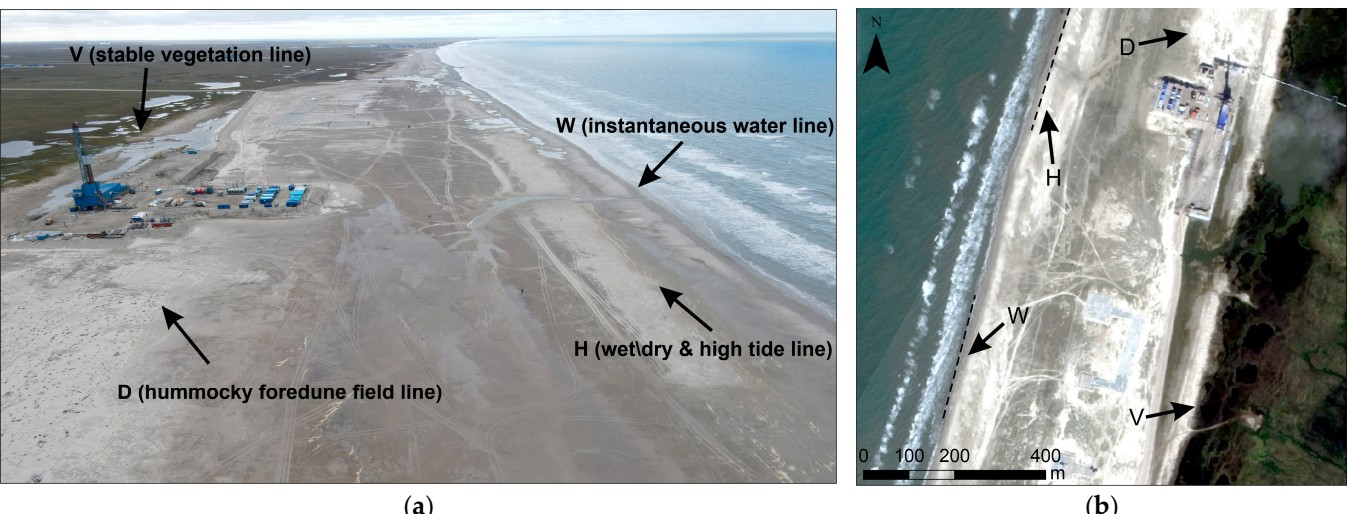

**Figure 5.** Detection of shoreline proxy on accretional coasts: (**a**) aerial photo (13 August 2022); (**b**) satellite imagery Jilin-1 GaoFen-03 (20 August 2022). Black arrows show different shoreline indicators.

In order to distinguish coastal segments with distinct decadal-scale shoreline behavior, we attempted a cluster analysis. A detailed explanation of this technique is beyond the scope of this article. The goal of cluster analysis is to organize big datasets into a smaller number of groups on the basis of their similarity [53,54]. In the present case, the main purpose was to detect the most common patterns of shoreline behavior among transect dataset. K-means clustering [55] with precluded numbers of clusters (3 and 5) was performed via Python script The proximity of the transects to each other is measured as the Euclidian distance between shoreline positions during each period. Clusters were named 3-1, 3-2, 3-3 for 3-cluster classifications and 5-1, 5-2, 5-3, 5-4, 5-5 for 5-cluster classifications, respectively.

### 3.3. Asessment of Geological Factors

To reveal the probable influence of different geological factors on spatial variability of coastal dynamics we analyzed a change in geomorphic, lithological and permafrost features along the coastline. Besides qualitative description, we attempted to perform quantitative correlative analysis for separate parameters, such as correlation between change in the rates of coastal retreat/advance and (a) width of beach and tidal flat for the entire coast, (b) height of bluff for the erosion coast, (c) steepness of beach slope for the accretional coast. These morphological parameters were calculated with the use of open-access ArcticDEM [36]. This DEM was constructed from stereoscopic pairs of Maxar satellite imageries, acquired between 2007 and 2022 over the summer seasons. The initial accuracy of provided strips is approximately 4 m in horizontal and vertical planes. The DEM was verified with the use of topographic map of 1:50,000 scale we have from archives, which allowed to enhance horizontal and vertical accuracy to approximately 1–2 m. Such accuracy is probably still not sufficient to calculate such parameters as beach slope. However, it could provide a general view of the studied topic.

For the sake of getting the beach widths, automated classification of the satellite image PlanetScope of 2020 was implemented, which allowed us to extract the area covering the beach surface. Then, the length of the polygon in every coastal transect was obtained. Bluff heights were extracted in the points of the beginning of every transect directly from the DEM. Beach slope was calculated for every transect as the relation of difference in elevation and distance between wet/dry line (H) point and instantaneous water line (W) point, which were digitized in the previous step of this study (Figure 5). Due to the poor

resolution of the input DEM obtained beach slope values are especially ambiguous in many places, though they present a general idea of variations in beach slope along the coast.

The cryolithological composition of the coast was extracted from the previous study, mainly from [19]. We distinguished four main types of the lithological composition of the bluff of the erosional coast: marine sands, lacustrine sands, marine and lacustrine sands and marine loams (Figure 3).

### 3.4. Hydrometeorological Forcing Assessment

3.4.1. Wave Action Assessment

The hydrometeorological forcing (or hydrometeorological potential—HMP) was determined as consisting of two parts [32]: (1) mechanical—the energy, coming to the coast from shoreward waves and called the wind-energetic potential of coastal dynamics; and (2) the thermal part—the energy transmitted to the coasts from the atmosphere (CD thermal potential). The wind–wave energy potential (WE) of coastal erosion was determined using an automated software Python package implementing the Popov–Sovershaev method [31,32]. Popov and Sovershaev put the wind–wave effect on coasts (in their works "wind–wave energy") in direct dependence on third power of the wind speed, duration of the ice-free period and wave acceleration length along the current wave direction, taken with the coefficient corresponding to wave height-wind speed dependence:

$$\mathrm{WE} = p \cdot 3 \times 10^{-6} V^3 x \cdot n \tag{1}$$

where V is the real wind speed of a chosen direction measured by anemometer at 10 m above sea level [m/s], x is wave fetch [km] along the current wind direction, $p$—the current wind speed and direction frequency during the ice-free period, $n$—ice-free period duration (days). The dimension of the $3 \times 10^{-6}$ coefficient according to [31] corresponds to the dimensions of $\rho/g$, where $\rho$ is density [t/m$^3$], and g is gravitational acceleration [m/s$^2$]. Thus, WE has the dimension of tons per year (if days in $n$ are calculated in seconds):

$$\frac{\mathrm{t}}{\mathrm{m}^3} \cdot \frac{\mathrm{s}^2}{\mathrm{m}} \cdot \frac{\mathrm{m}^3}{\mathrm{s}^3} \cdot \mathrm{m} \cdot s = \frac{\mathrm{t}}{\mathrm{s}} \cdot s = t \tag{2}$$

The total year value of wind–wave impact is defined as the sum of particular values determined for all wave-prone directions and wind speeds exceeding 5 m/s. Considering the dimension of the WE the measure of "wind–wave energy" impact proposed by Popov and Sovershaev, this technique actually denotes the mass of water coming per meter of the wave front during the ice-free period. Since the dimension of this value does not correspond in the physical sense to the term "energy", a previous study [32] proposed the term "wind–wave energy potential" as a certain energy characteristic. It is accumulative (during the ice-free period) and can be consumed by the coast or not depending on the input conditions (the coast ice content and particle size, the height of the cliff and other endogenous factors).

The input data for these calculations are bathymetry and wave acceleration data along wave-prone directions, determined using a digital elevation model [56]. The surface (10 m above sea level) wind speed and directions were obtained from the ERA-5 reanalysis data [57] of the European Center for Medium-Range Forecasts ECMWF and the MERRA2 reanalysis data [58] of the US Aerospace Agency NASA. The ice-free period characteristics were derived from sea ice concentration satellite data from the US National Ice and Snow Center, NSIDC, Climate Data Records [59]. The identification of the ice-free period was carried out using a modified threshold method adapted for the Arctic coastal zone data [42,60]. The start/end dates of the ice-free period are presented as integers corresponding to the number of the day of the year, starting from January the 1st. The duration of the ice-free period is estimated in days. Long-term trends in the characteristics of the ice-free period are given in units of "days/10 years".

To evaluate the Popov–Sovershaev method, which is simple and carries an empirical wind–wave relationship and actually expresses the mass of water, coming to the shoreline, the direct calculations of wave energy (in Wt/m) by the spectral wave model WAVEWATCH III (WWIII, version 6.07 [30]) were conducted. This model is based on the numerical solution of the equation of the wave action density spectrum taking into account effects of the energy dissipation, non-linear interactions, and bottom friction and involves wind speed, ice concentration data. The modern ST6 scheme of WWIII [61,62] was used as it was previously shown that it provides the best results [63,64] if compared to buoy and satellite sea wave height data.

Wind speed on a 10 m above the ground and sea ice concentration data for the wave model input data were taken from the NCEP/CFSR reanalysis (1979–2010) with spatial resolution of approximately ~0.3° [65] and NCEP/CFSv2 reanalysis (2011–2021) with resolution of ~0.2° [66], and temporal resolution of 1 h. The reanalysis data were downloaded from the server [67]. Linear interpolation of the reanalysis data to the unstructured mesh was performed by using Python code. The mesh is dense in the coastal region under investigation and loose in the surrounding area: the resolution is 10 km in the open sea and 700 m nearshore. The bathymetric data were obtained from the ETOPO 1–min bottom topography database [68] and detailed navigation maps. This grid covers the Barents Sea, the Kara Sea, and the entire northern part of the Atlantic Ocean. A more detailed description of the model configuration, the main features of the experiments with the unstructured mesh and quality assessments for different regions are presented in [64,69–72]. The wind wave characteristic fields (3-h data for the ice-free period) from 1979 to 2021 (total of 42 years) were got as model output. These include the sea wave height, wave length, wave energy and some others.

### 3.4.2. Thermal Action Assessment

The thermal potential (TE) of thermal denudation is estimated by the air thawing index showing the number of positive °C-days per year [73]. A similar parameter called degree-days thawing was used in [9]. This index is the evaluation of the annual amount of heat added to the ground and permafrost during a warm period. ERA5 reanalysis 2 m above surface temperature data were used [57]. The previous research [32] showed that the ERA reanalysis correlates to observation data in the region with a coefficient of 0.87–0.96 and RMSE amounts to 8–12%.

Long-term trends were calculated based on a linear fit of the series by the least squares method. The decision on the statistical significance of trends was taken at a significance level of 0.01 using Student's *t*-test.

## 4. Results

### *4.1. Coastal Dynamics*

#### 4.1.1. Multi-Decadal Shoreline Change Rates

At a multi-decadal timescale (1964–2022), ice-rich permafrost bluffs exhibited a tendency towards persistent erosion (Table 2; Figure 6a). The average shoreline change rate for erosional stretch is −1.03 m/yr and varies significantly (standard deviation is 0.66 m/yr) alongshore. The southwestern part of this stretch (transects 1–60) is a robust erosional hotspot, the mean annual retreat is up to −2.75 m/yr (transects 44–51). Coastal segments northward to Nyavotolovayakha River had reduced erodibility, with a mean recession rate of less than 1.0 m/yr. The most stable (approximately −0.1 m/yr) coastline stretch is located at the southwestern border of the Kharasavey settlement (transects 150–160). Technogenically transformed reaches in the alignment of the settlement (transects 169–195) have faced moderate erosion, with a mean and maximum change rate of −0.93 m/yr and −1.37 m/yr, respectively.

**Table 2.** Shoreline change rates (m/yr) of the erosional coasts.

| Time Period | 1964–1972 | 1972–1977 | 1977–1982 | 1982–1988 | 1988–2006 | 2006–2014 | 2014–2016 | 2016–2020 | 2020–2022 | All Period (1964–2022) |
|---|---|---|---|---|---|---|---|---|---|---|
| Mean rate | −1.08 | −1.02 | −1.29 | −1.49 | −0.85 | −1.12 | −1.07 | −0.60 | −0.84 | −1.03 |
| Standard deviation | 0.83 | 1.16 | 1.40 | 1.79 | 0.70 | 0.83 | 0.86 | 0.61 | 0.77 | 0.66 |
| Peak rate | −4.27 | −5.68 | −6.62 | −7.86 | −2.52 | −4.34 | −4.01 | −3.04 | −4.08 | −2.75 |

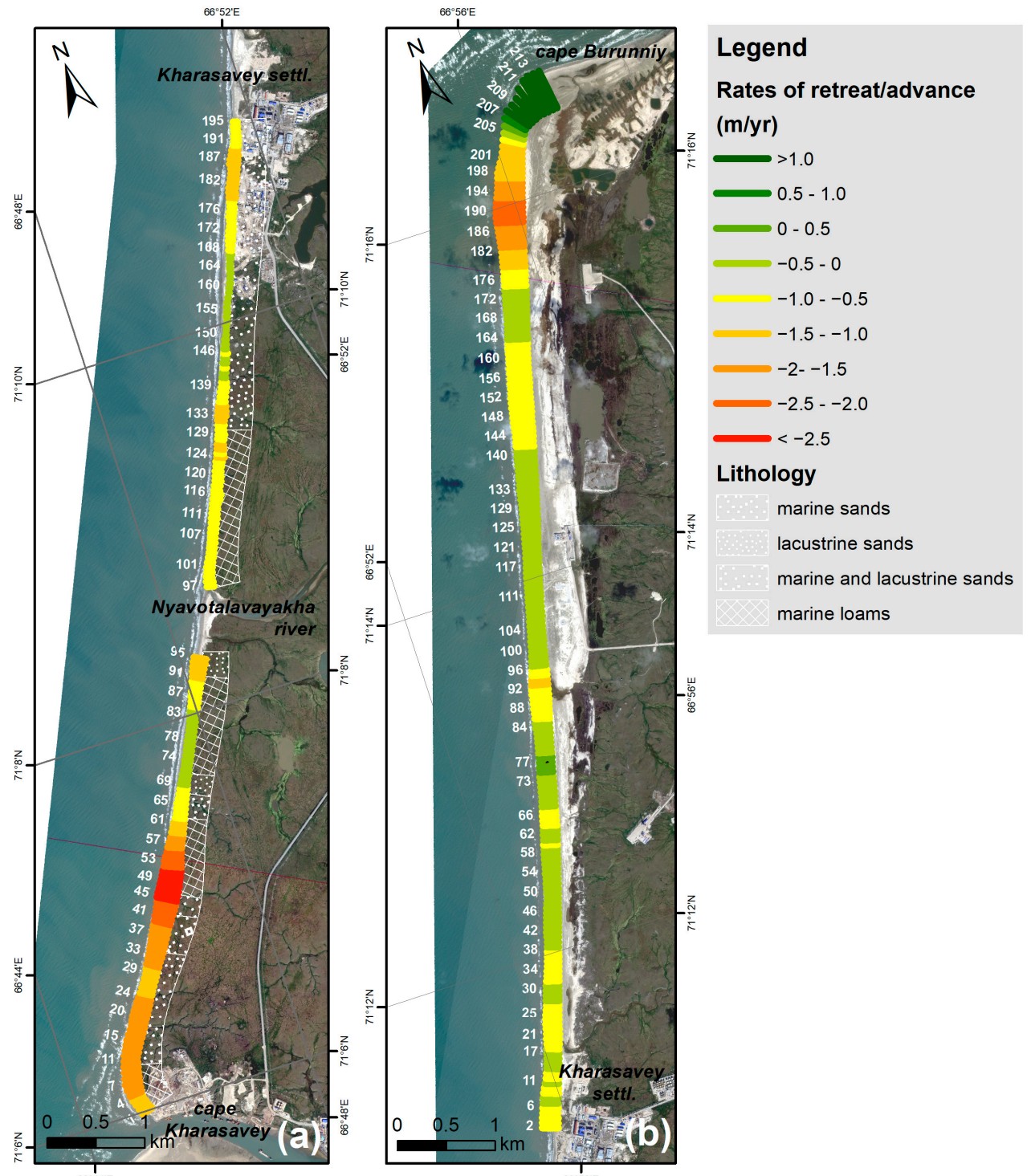

**Figure 6.** Multi-decadal (1964–2022) coastal dynamics of the erosional (**a**) and the accretional (**b**) stretches. Background is Jilin-1 satellite image (2022).

In contrast, shoreline change analysis along accretional coasts revealed both advancing and retreating reaches, 6% and 94% of the coastline, respectively. The average multi-decadal change rate is −0.42 m/yr (Table 3) and has substantial spatial variability (standard deviation is 0.99 m/yr). Hence, the majority of transects of accretional stretch showed slightly retreating values (Figure 6b). Within this stretch, a slight erosional gradient (towards Cape Burunniy) exists as a probable consequence of longshore drift. The highest rate of retreat (up to −2.03 m/yr) at the accretional coast is evident in 1 km southward to Cape Burunniy (transects 188–192). Persistent shoreline progradation has occurred locally in the alignment of the cape (transects 208–214). The highest seaward movement of the shoreline (up to 5.1 m/yr) is probably associated with the northeastern spit aggradation.

**Table 3.** Shoreline change rates (m/yr) of the accretional coasts.

| Time Period | 1964–1977 | 1977–1982 | 1982–1988 | 1988–2006 | 2006–2014 | 2014–2022 | All Period (1964–2022) |
|---|---|---|---|---|---|---|---|
| Mean rate | 1.70 | −1.66 | 0.25 | −3.24 | 3.76 | −1.43 | −0.42 |
| Standard deviation | 1.33 | 2.27 | 4.79 | 2.14 | 4.59 | 1.83 | 0.99 |
| Min rate | −5.03 | −5.86 | −8.99 | −7.51 | −0.84 | −6.43 | −2.03 |
| Max rate | 5.77 | 6.34 | 25.11 | 0.80 | 34.12 | 2.99 | 5.12 |

4.1.2. Spatiotemporal Variability of Coastal Dynamics

The shoreline of the studied area showed substantial changes in the rates of its movement with time on both erosional and accretional types of the coasts. On the erosional coast, a slight ($R^2$ = 0.2842) trend to increase rates values, meaning a decrease in coastal erosion, is observed (Figure 7a). Dynamics of the accretional coast express a very slight ($R^2$ = 0.0037) opposite trend that is due to non-linear, cyclical shoreline behavior (Figure 7b).

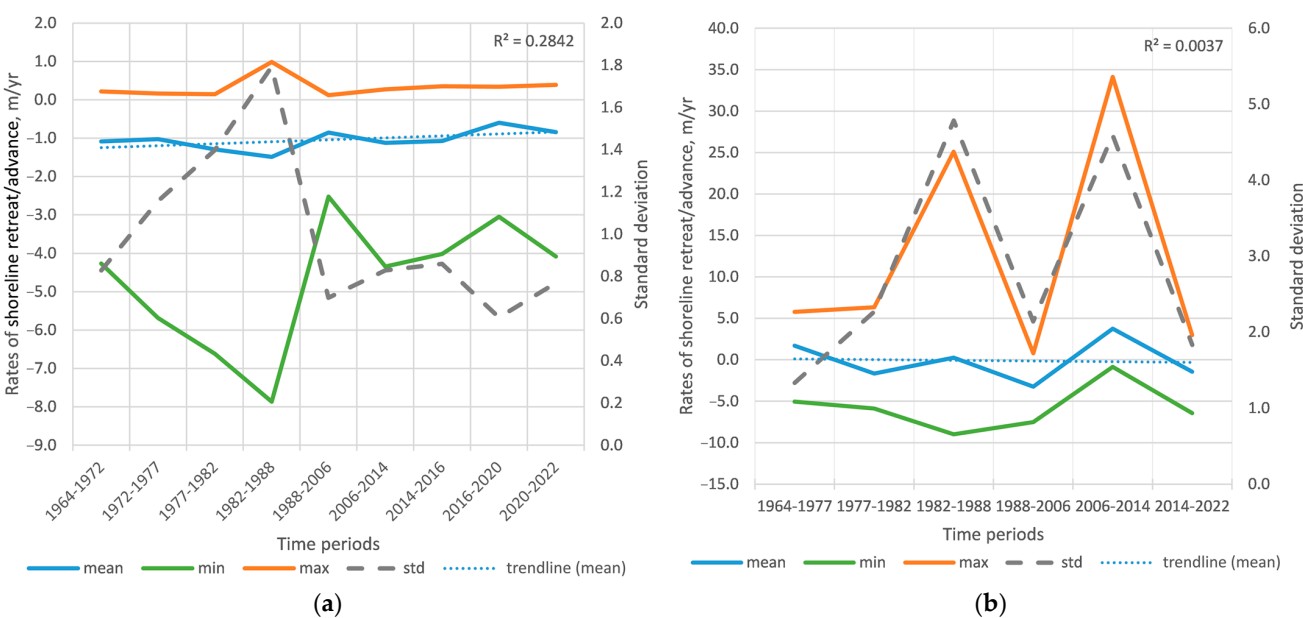

**Figure 7.** Rates of shoreline retreat or advance for different periods of the erosional (**a**) and the accretional (**b**) coasts.

The erosional stretch (Figure 8a) faced the most considerable erosion during 1982–1988 (mean rate was −1.5 m/yr, peak rates are up to −7.9 m/yr). The most stable period was 2016–2020 (mean rates were −0.6 m/yr). Rates of shoreline movement of accretional coasts (Figure 8b) vary much more both in spatial and temporal scales (std ranges from 1.3 to 4.8 m/yr for different time periods) (Table 3). Mean rate changes from negative to positive values with time. It was the highest (mean rate 3.8 m/yr, maximal rate 34.1 m/yr) in

2006–2014, and it was the lowest (−3.2 m/yr mean) in the previous 1988–2006 period (the absolute minimum −9.0 m/yr was reached in 1982–1988).

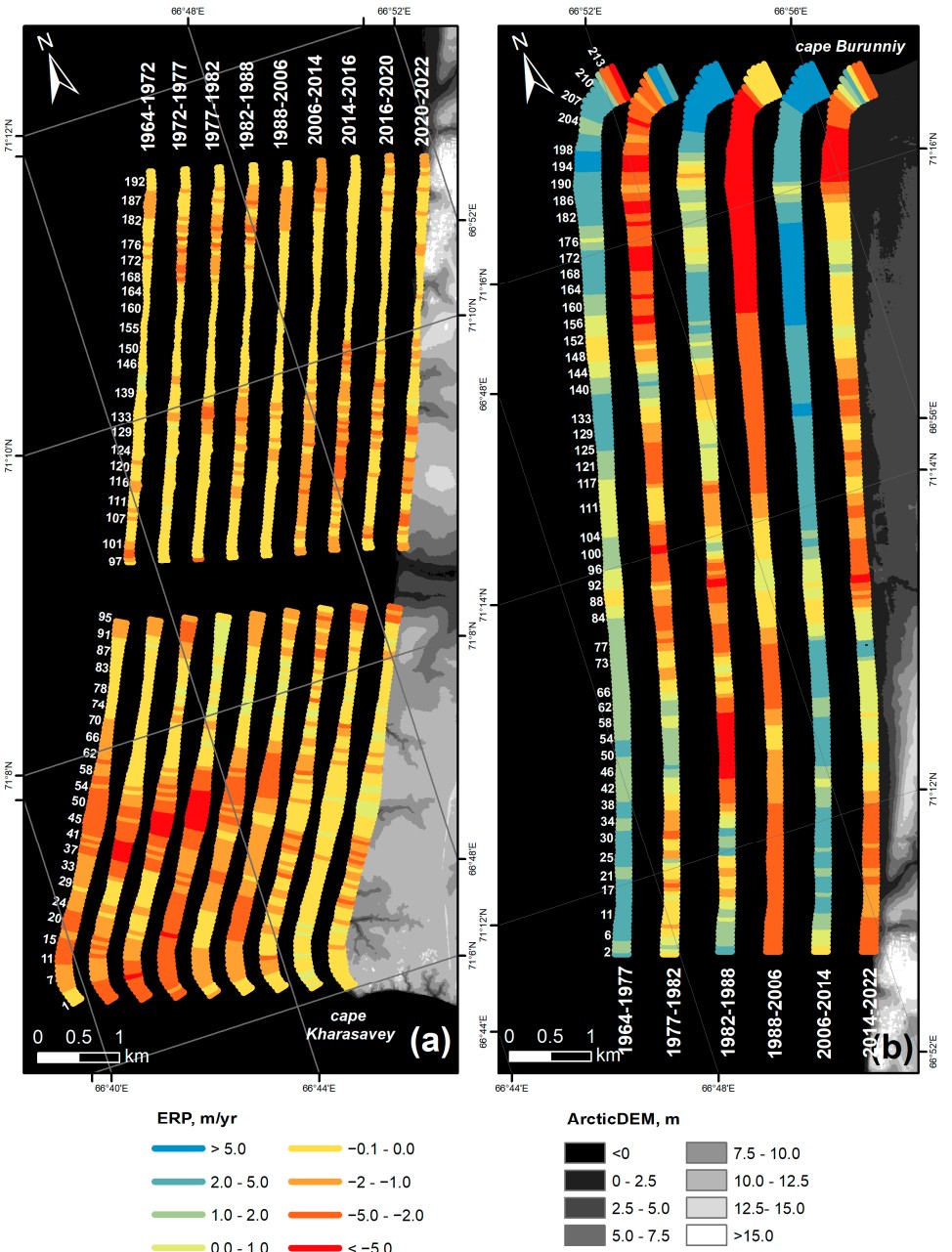

**Figure 8.** End point rates (EPR) of the erosional (**a**) and accretional (**b**) coasts for different periods. Background is ArcticDEM [36].

At the beginning of the studied period (1964–1972), the shoreline change rates of erosional stretch were close to the mean rate of retreat for the entire period of study (−1.1 m/yr). Erosion with a rate of up to −4.3 m/yr took place mainly in the southwestern part of the area, while north-eastern part was more stable. In 1972–1977, the mean rate stayed approximately the same (−1.0 m/yr), though the intensification of erosion (up to −5.7 m/yr) at the central part of SW section (transects 35–45) occurred. The same period the erosion ranged of −1 m/yr to −3 m/yr started at the northern part of the section in the area of the settlement that just started to be built (1976). In the next period (1977–1982), the rates of retreat was increasing (−1.3 m/yr on average, −6.6 m/yr—maximal retreat), as well as the length of the coast affected by erosion. Two new areas with activated erosion

appeared: on the south at Kharasavey Cape (transects 1–8) and in the middle of the area (transects 128–135). In the following period (1982–1988), retreat of coastal bluff in the area reached its maximum (−1.5 on average, −7.9 maximum), with the rates exceeding 5 m in two sites of the SW part of the coast.

In the northern part in the alignment of the Kharasavey settlement being under construction the coast retreated quite actively as well (−0.8−−1.4 m/yr). The following 1988–2006 years were among the calmest (mean is −0.9 m/yr), though the center of erosion in SW part (transects 44–57) stayed quite active (−1.5−−2.7 m/yr). In the following periods (2006–2014, 2014–2016, 2016–2020, 2020–2022), erosional process became calmer (mean rates were −1.1, −1.1, −0.6, −0.8 m/yr, respectively), in particular SW erosion center, where by the 2016–2020 rates of retreat dropped by around 0.5 m/yr. In 2006–2014, a new center of erosion in the central part of erosional coast (transects 110–130) appeared. The coastal erosion was 1–2 m/yr retreat in this period, and it intensified during the next period by 2–3 (up to 4) m/yr retreat. In the period 2020–2022, erosion is still quite active in this area (up to −4.1 m/yr) (Figure 8a).

In contrast, during the first period of 1964–1977 the accretional shoreline was mainly advancing (average rate amounted 1.7 m/yr), especially in the most northern (transects 190–200, it had maximal rates of advance—up to 5.8 m/yr) and most southern parts (smaller—up to 3.4 m/yr). A small section at the most north of Cape Burunniy was retreating rapidly (up to −5 m/yr). In the next period, 1977–1982 the shoreline experienced mostly negative dynamic (−1.7 m/yr on average), except separate sections with slight (0.5–1.0 m/yr) advance on the south and in the center of the accretional coast, and intensive expansion of the spit of Burunniy Cape (up to 5.9 m/yr). 1982–1988 was the most contrast regarding the coastal dynamics (the same situation was observed on the erosional coast). The average rate was slightly positive (0.3 m/yr), though the rates were substantially changing along the coastline, reaching its maximum (up to 25.1 m/yr) at the spit of Cape Burunniy, and its minimum (up to −9.0 m/yr) in the central part of the area (transects 43-58) (Figure 8b). During 1988–2006, the coastal dynamic was very uniform along the coast, mostly retreating, with medium rates (3.2 m/yr on average). During the following 2006–2014 it was in opposite quite uneven, mostly advancing and with high rates (3.8 m/yr in average, 34.1 m/yr in maximum). A substantial shoreline progradation took place in the neighborhood of Cape Burunniy and in almost the entire NE part of the coast (transects 110–185). In the last period (2014–2022), the coastal dynamic was quite calm: with a general retreating pattern (mean rate was −1.4 m/yr), but with medium (1–2 m/yr) accretion at several sites.

*4.2. Cluster-Based Classification of Patterns of Shoreline Behavior*

Cluster analysis of the erosional stretch (Figure 9a) reveals the prior, most widespread cluster (3-1, 71% of the coastline). Cluster 3-1 comprise the regional-scale pattern of coastal behavior with a clear erosional signature of subtle temporal variation. The average shoreline change rate along this cluster is approximately 0.8–1.1 m/yr for each period. Thus, it varies insignificantly and shows a slight tendency towards intensification of coastal erosion. Other (3-2 and 3-3) less populated clusters located on the south of the area (Figure 9a) clusters represent patterns of behavior more similar to the change in minimal rates of shoreline movement with severe intensification of erosion in the 1970–1980s and significant attenuation of coastal processes after 2014 (Figure 10a). Distinguishing of five clusters (Figure 9b) does not provide an identification of something conceptually new: the first cluster (3-1) from the previous clustering is subdivided into two approximately similar (5-1 and 5-2). Another 3 clusters (5-3, 5-4, 5-5) are close to the previous 3-2 and 3-3, though the 5-5 cluster (5% of all transects) differs by shifting of the erosional peak to (Figure 10c). Hence, cluster analysis identified outliers of the southwestern part of the erosional stretch. Local-scale, non-climatic drivers exerted great rapidity of erosion among these reaches.

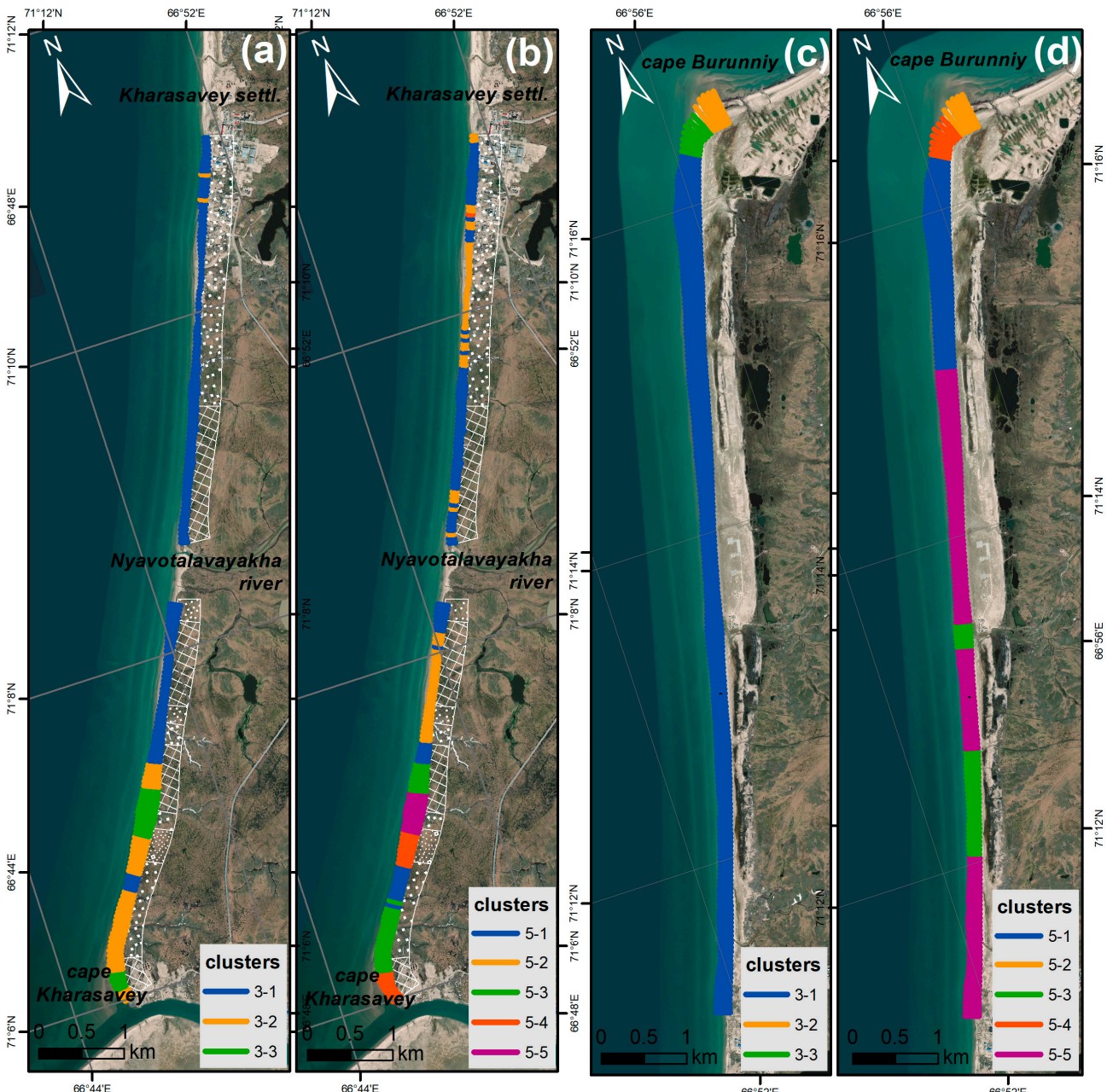

**Figure 9.** K-means clustering of erosional (**a**,**b**) and accretional (**c**,**d**) coasts. A, C—3 clusters distinguished, B, D—5 clusters distinguished. Background is GeoEye-1 (2014).

At the accretional stretch, 3-clustered classification is clearly insufficient to describe all patterns of coastal behavior, as far as shoreline movements at Burunniy Cape are very different from all the other reaches (Figure 9c). With this classification, two of three clusters (3-2 and 3-3) were distinguished at this Cape, whereas all the rest of the coast (94%) was referred to the first (3-1) cluster. This first cluster expresses the pattern close to the pattern of the first cluster of the erosional coast but reversed (Figure 10b). Clusters 3-2 and 3-3 represent the peaks of coastal retreat and advance replacing one another with time but with their different amplitudes. Generation of the 5 clusters (Figure 9d) allowed to distinguish 3 more or less different clusters (5-1, 5-3 and 5-5) within the framework of the first cluster of the previous clustering (3-1). The most different among them is 5-3 cluster having a peak of erosion in 1982–1988, whereas all the other clusters have a peak of accretion (Figure 10d). Cluster-based segmentation of the accretional coasts captured a slight response of wave-

impact intensification and subsequent reversal of sediment fluxes that may be exerted by drastic changes in sea-ice extent [42].

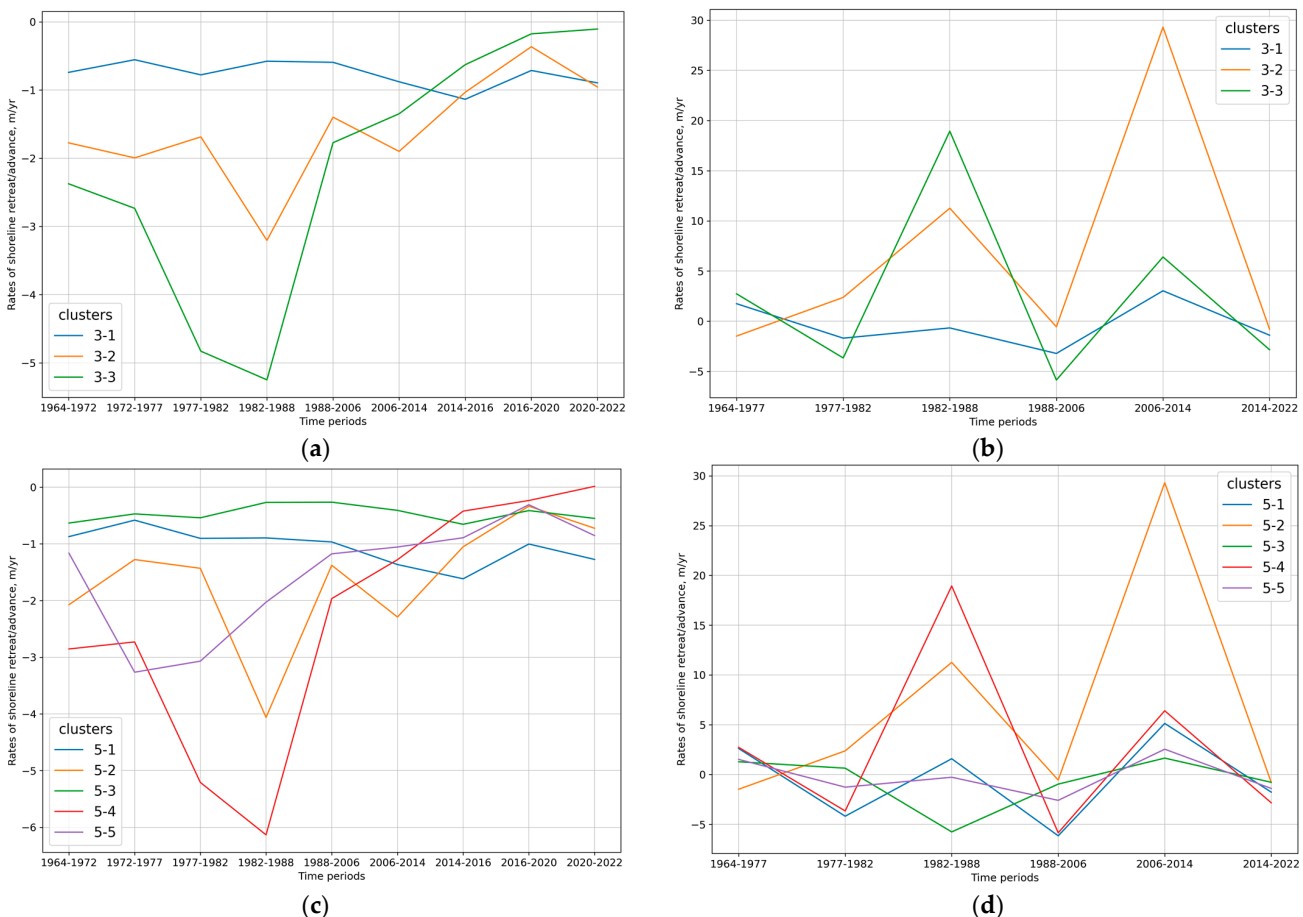

**Figure 10.** Change in mean rates of retreat/advance for the obtained clusters: (**a**) for the erosional coast, 3 clusters distinguished; (**b**) for the erosional coast, 5 clusters distinguished; (**c**) for the accretional coast, 3 clusters distinguished; (**d**) for the accretional coast, 5 clusters distinguished.

### 4.3. Geomorphic and Cryolithological Features

Geomorphic features were estimated for both erosional and accretional stretches (Figures 11 and 12) via remote sensing data in order to compare with spatiotemporal variability of shoreline change rates (Figures 11a and 12a) and mean end point rates (Figures 11b and 12b):

- **Cryolithology.** At the erosional part of the coast, we distinguished and mapped four principal types of sediments composing the cliffs: marine sands, lacustrine sands, marine and lacustrine sands and marine loams (Figure 11e).
- **Beach width.** The width of the beach has an average value 50 m at the erosional part of the coast and approximately 234 m at the accretional stretch (Figures 11c and 12c). It varies significantly, especially at the accretional coast (standard deviation is 20 m at the erosional coast and 82 m at the accretional). It reaches its maximal value of 511 m at the NE in the area of Burunniy Cape, and its minimal value of 20 m at the SW part (32 transect). In general, the width of the beach is increasing towards the capes, especially at the SE of the accretional stretch. Correlation coefficients between beach width and rates of retreat/advance values are −0.267 for erosional coasts and 0.458 for accretional coasts.
- **Bluff height.** The height of the cliff was measured for the erosional part of the coast (Figure 12d). It amounts to 7.5 m in average, but it varies slightly (std = 1.9 m) along

the coast. It is minimal in river valleys (up to 2.3 m), and it is maximal (13.2 m) in the NE part (transect 168). The correlation coefficient between cliff height and rates of retreat/advance values of erosional coasts is −0.052.

- **Beach slope**. Slope of the beach was calculated for the accretional stretch of the coast (Figure 12d). It has an average value 0.002, which varies substantially (std 0.017). It is maximal (0.040) in the SW part of the section (transect 30). It's minimal (−0.004) in the same area (transect 20) and in some other areas of the coast (around −0.03 in the areas of transects 100 and 190). The values are not precise, though they present the general view. Correlation coefficient between beach slope and rates of retreat/advance values of accretional coasts is −0.063.

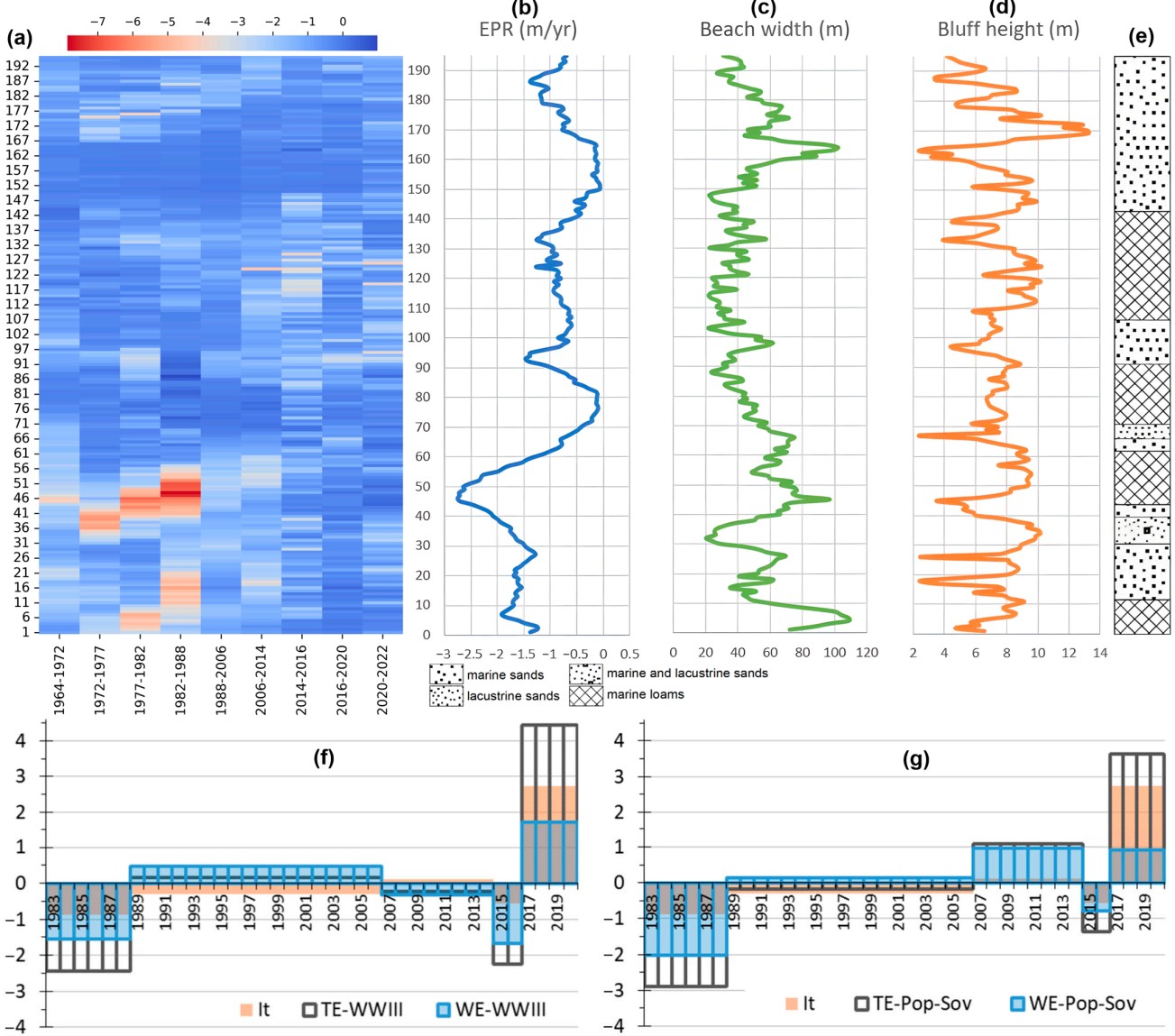

**Figure 11.** Environmental drivers determining coastal behavior of the erosional stretch: (**a**) change in rates of retreat/advance (m/yr) for different periods along the transects; (**b**) change in mean (1964–2022) rates of retreat/advance (m/yr) along the transects; (**c**) change in beach width (m) along the transects; (**d**) change in bluff heights (m) along the transects; (**e**) change in lithology along the transects; (**f**) HMP averaged for different periods according to the WW-III model; (**g**) HMP averaged for different periods according to the Popov–Sovershaev technique.

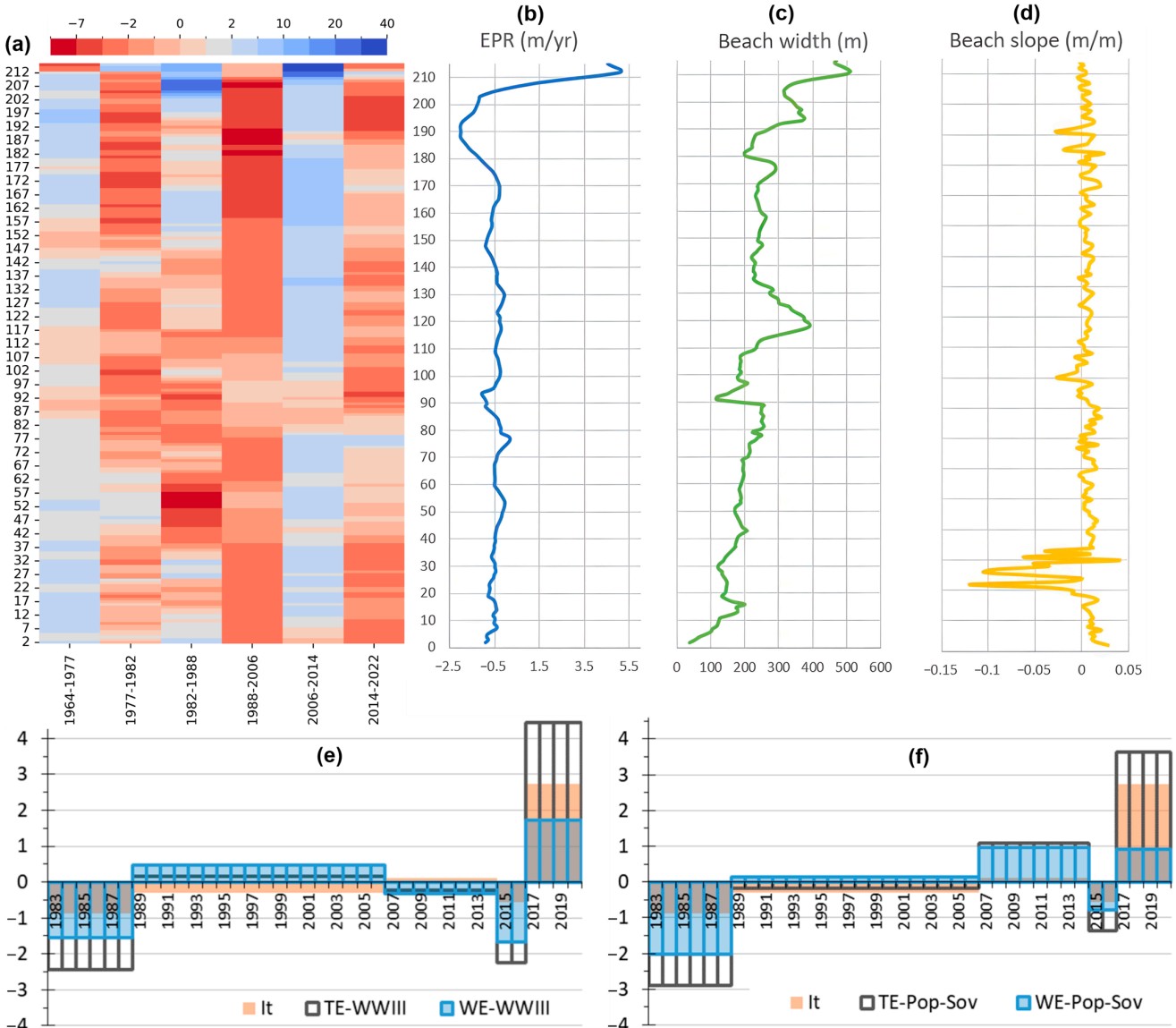

**Figure 12.** Environmental drivers determining coastal behavior of the accretional stretch: (**a**) change in rates of retreat/advance (m/yr) for different periods along the transects; (**b**) change in mean (1964–2022) rates of retreat/advance (m/yr) along the transects; (**c**) change in beach width (m) along the transects; (**d**) change in the beach slope (m) along the transects; (**e**) HMP averaged for different periods according to the WW-III model; (**f**) HMP averaged for different periods according to the Popov–Sovershaev technique.

### 4.4. Hydrometeorological Forcing

In general, the dynamic of the WWIII wave energy is consistent with the wave energy potential calculated by Popov–Sovershaev's method (Figure 13a,b). The main features are:

- The main peaks coincide (especially high values in 1994, 2007, 2020 and low in 1979, 1981, 1984, 1992, 1998, 2004, 2013, 2019);
- Periods of relatively high (1994–1997, 2005–2012, 2016–2021) and low (1979−84, 1998–2003, 2013−15) values;
- Both series are characterized by high within-sample variability (for WWII it is 33% of mean value, for Popov–Sovershaev's WE-potential—50%);
- Both polylines have positive and statistically significant trends, which means that wave energy at Kharasavey area has increased since the beginning of the 1980s.

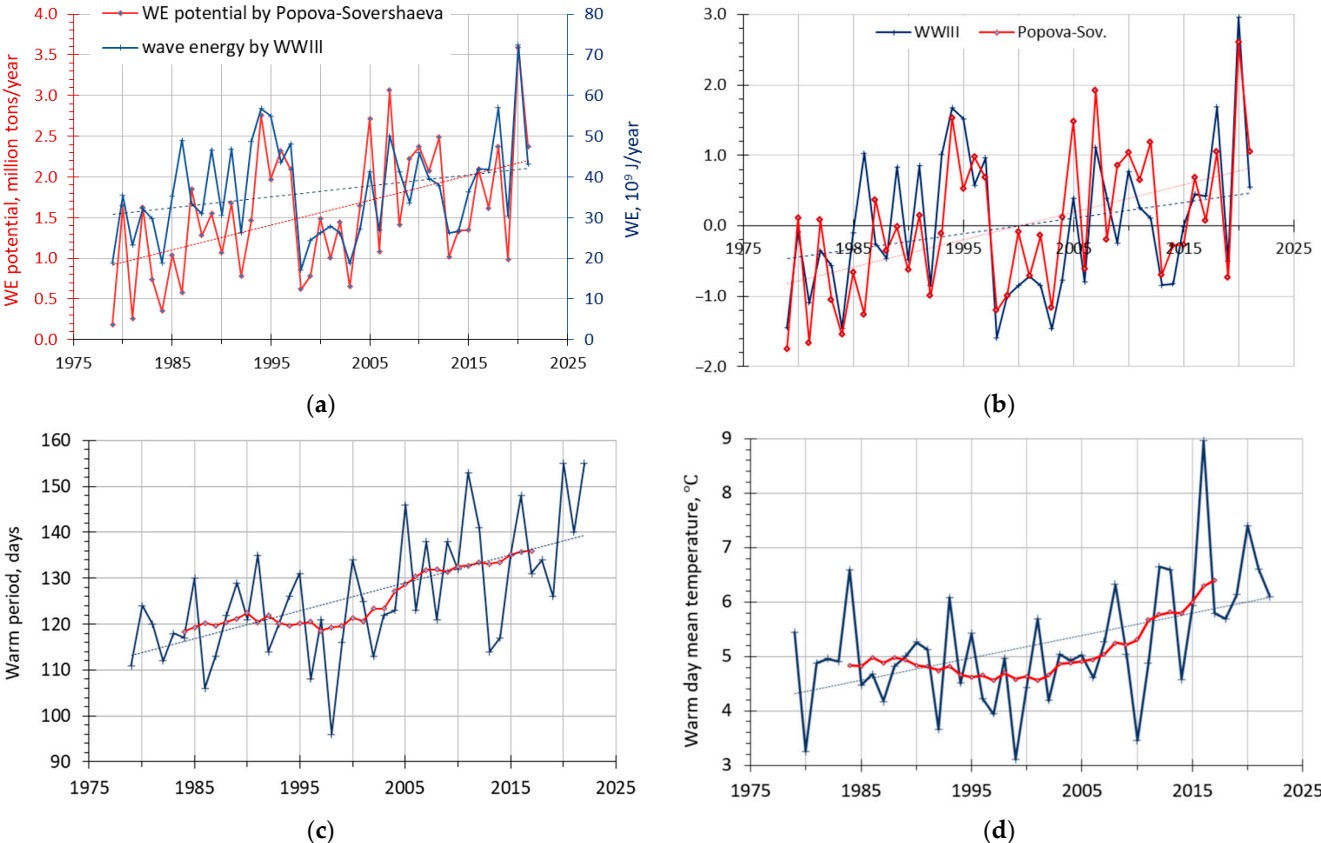

**Figure 13.** Hydrometeorological forcing at the Kharasavey area: (**a**) Wind–wave energy assessment at Harasavey by Wave-WatchIII (WWIII) model (right blue axis) and wind–wave energy potential by the Popov–Sovershaev method (left red axis) in absolute values and their linear trends; (**b**) the same expressed in anomalies normalized by standard deviation; air thawing index and its linear trend and 11-year running mean (**c**) with a number of warm days (warm period, days); (**d**) with a mean daily temperature of warm days.

The main difference is in trend coefficient: if compared to the period mean WWIII has 0.7%/year or 32% per 43 years, which makes up 96% of series standard deviation. For WE-potential, the figures are higher: ~2%/year or 84% of sample mean per 43 years, which is 170% of standard deviation. The difference occurs due to lower WE-potential values in 1980s, and higher in 2000s. The correlation coefficient of WWIII and Popov–Sovershaev's series is 0.79.

Air thawing index (Figure 13c,d) has increased by 290 °C-days since 1979, which is close to the half-mean value (that is approximately 630 °C-days) and exceeds the series' standard deviation (that one is 180 °C-days) by 1.6 fold. It occurred due to both warm period extension and mean warm-day temperature increase.

The warm period has increased by approximately 25 days, which is 20% of long-term mean or 190% of series' standard deviation (which is 13 days). The mean warm day has become warmer by approximately 1.8 °C per 41 years (from 4.3 up to 6.1 °C) which is 35% of long-term mean (or 150% of standard deviation which is 1.1 °C). After 2006 the mean warm daily temperature has never dropped below the long-term mean. There are also extremely warm years of 2016, which is 2–3-fold higher in terms of Iat than long-term mean value, and also 2012 and 2021.

## 5. Discussion

The obtained results illustrate a clear tendency towards persistent erosion along most of the studied stretches, both accretional and erosional. Previous researches of the Kharasavey key site were focused on the dynamics of the erosional stretch and had a

restricted spatial coverage [19,20]. Based on these studies, a significant increase in erosion rates over the past decades was established. Although the mean erosion rates show little difference from previously identified, the current results of intra-decadal shoreline change rates partially challenge this conclusion—the abrupt intensification of coastal erosion does not represent a regional tendency but is rather local-scaled. Through cross-proxy analysis, it was feasible to, on one hand, discern a subtle climatically predetermined pattern and, on the other, exclude extreme values and endeavor to discuss the determinants underpinning them.

### 5.1. Spatial and Temporal Patterns of Coastal Dynamics

Rates of shoreline movements in the studied area are higher than the long-term annual average rates for the entire Arctic (0.5–0.6 m/yr), and for the Kara Sea (0.7 m/yr) [3]. However, in general, the dynamic of the studied coast is rather typical for the similar ice-rich permafrost bluffs and accretional coasts of the Kara Sea. For example, human-affected ice-rich permafrost bluffs of Ural site of the Baydara Bay have a very similar set of factors: it is also rather straightened, exposed to the waves of open sea, composed of sands and loams with high ice content or containing massive ice beds/wedge ice, have similar height of bluff and width of the beach. During a similar period (1964–2016), it retreated with a comparable mean rate of approximately 1.2 m/yr [11]. Close values were recorded for the similar period at the eastern coast of Beliy island (1.2 m/yr in average) [74]. The highest rates of erosion for the Kara Sea region were observed at Marre-Sale (2 m/yr on average for 1969–2009) [10] and at the western reaches of Beliy island (1.9 m/yr for 1969–2016) [74], though these coasts differ in cryolithological composition (high ice content, massive ice beds inclusions at Marre-Sale) and geomorphic features (low cliffs, exposition to stronger open sea waves at the west of Beliy island), making them more vulnerable to coastal thermal erosion.

Long-term annual tendency to retreat of the accretional coasts is evident in many sites of the Kara Sea: accretional reaches of Ural coast of Baydara Bay are retreating with a mean rate of 1.7 m/yr (1964–2016) [11], low-lying sandy coast of Gydan peninsula in the area of Salman gas field is eroded by 1.1 m/yr (1972–2020) [75]. It is common in other regions as well: in the Canadian Arctic [51,76] and USA [12,77]. Spatial distribution of the shoreline movements is rather typical at the studied site: rates of shoreline change are higher when closer to the capes and at the areas of significant technogenic pressure, and lower (or advancing) in the areas less exposed to sea waves and in proximity to the river mouths. However, the temporal pattern of the shoreline change is somewhat different from other reported studies. In recent decades, most of the coasts of the Kara Sea, especially permafrost bluffs, showed a tendency to acceleration of erosion after the increase in hydrometeorological pressing induced by climate change [11,41,78]. It is reported for the other regions of the Arctic as well [6,7,12,79,80].

In the present study, we do not observe any clear tendency to either intensification or attenuation of coastal erosion in a long-term prospect (Tables 2 and 3). It may be related to an extremely rapid shoreline collapse in the 1970–1980s, which is not common in other sites of study and not provoked by hydrometeorological factors, but probably by the beginning of construction in the area. Alternatively, the coast may be less sensitive to the changing climatic conditions due to its composition, excessive sediment balance or shoreline management.

### 5.2. Environmental Drivers Determining Spatial Variability of Coastal Dynamics

It might be tempting to link differences in shoreline change rates with geomorphic and cryolithological features of the area. Non-linear interaction of sediment supply, thermal abrasion and thermal denudation may lead to the exclusion of marked correlation between morphological or cryolithological features and coastal dynamics:

- **Cryolithology.** Lithological (grain size, consolidation, texture) and permafrost features (ice content, tabular/wedge ice presence) of coastal zone play an essential role in its

dynamic, especially for permafrost coasts [26]. The cryolithological composition of the bluff varies substantially along the studied erosional coast (Figure 11e). Marine and lacustrine sands mainly have low (<30%) ice content, whereas marine loams have high (>40%) ice content [19]. Comparing the rates of retreat averaged by different types of lithology, the mostly eroded were the coasts, composed of a combination of marine and lacustrine sands (−1.9 m/yr on average). It is a small (450 m length) section of the coast between transects 35 and 41, which is why it may be confusing to compare it with larger areas of other lithological types. Transects with marine loams retreat with an average rate of 1.0 m/yr, that is slightly faster than transects with lacustrine and marine (separately) sands (−0.9 m/yr for every type). The coastal section in the SW part (45–51 transects) retreated the most considerably (Figure 6a) and is composed of marine loams with high ice content. Thus, neither average retreat rates are approximately similar, it was noted in the previous publications on this area [19,20], that this section is more prone to erosion. Lower resistivity to coastal erosion of fine-grained sediments (loams, silts, and fine sands), especially in permafrost environments (as containing more ice), compared to medium and coarse-grained sands was also determined in the other studies before [81,82].

- **Beach width.** One of the most important parameters influencing coastal erosion is the width of the beach, as the beach protects the coastal bluff from wave erosion. It was demonstrated for the Ural coast of Baydara Bay: after the beach width reduction with time, the rates of coastal retreat increased [11]. The correlation of beach width and the rates of retreat for both erosional and accretional coasts is extremely low (−0.267 and 0.458 for erosional and accretional stretch, respectively), that may be related to the ambiguity of beach extraction from imagery, or may indicate low significance of this parameter for coastal processes in the area (Figure 12c). However, at least at the accretional coast, we observe that as a general tendency, the higher beach corresponds to the decreased retreat rate, probably, due to the positive sediment supply.

- **Bluff height.** Theoretically, low coasts are more susceptible to coastal erosion than high ones, as they demand less wave energy to be eroded than the high ones, however, it does not always work [83,84]. The calculated coefficient of correlation between coastal bluff height and the rates of coastal retreat along the erosional stretch is negligible (−0.051), that may be explained by poor accuracy and reliability of the ArcticDEM that was used for calculation and due to predominance of other factors. Previous study [85] determined that at the Ural coast of Baydara Bay offshore depths and slope are the leading drivers of coastal retreat, while bluff height is of subordinate importance. Moreover, the change in bluff height along the studied section of the coast (Figure 11d) is not very substantial (std = 1.9 m), that is why in particular this factor apparently is not very crucial here.

- **Beach slope.** A steep slope of the beach may be both an indicator and/or a consequence of an intensive coastal erosion. The available DEM does not allow building any reliable correlation in the coastal zone: coefficient of correlation between the calculated rates of retreat/advance and beach slope for accretional stretch is −0.062. However, we see a decrease in the beach slope at the NE on Burunniy Cape, which correlates with an increase in the beach width, and with attenuation of coastal retreat, but not for every time period (Figure 12d).

It may be declared that the accuracy of the geomorphic features, afforded by open-access remote data, is insufficient to highlight and accentuate patterns of coastal change. In terms of quantifying the interplay of coastal behavior and onshore/offshore morphology, local-scale field measurements are appropriate. Further understanding temporal patterns of shoreline behavior and its dependence on geological features also requires identifying local-scale differences in wave impact, which may be derived by upscaling of the applied models.

*5.3. Environmental Drivers Determining Temporal Variability of Coastal Behavior*

Temporal variability of coastal dynamics is mainly determined by hydrometeorological parameters, such as change in directions and speeds of winds generating geomorphologically significant wave-events, sea level fluctuations (long-term rise and short-term tides and surges), the ice-free period and amount of days with positive temperatures per year [1]. However, this study revealed little intra-decadal coherence of shoreline change rates and calculated hydrometeorological effect trends via the Popov–Sovershaev method (Figures 11f and 12e) and WaveWatch III (Figures 11g and 12f). Neither maximums, nor minimums of both wind–wave and thermal parameters do not correspond to the intensification of coastal erosion on intra-decadal scale. However, some clusters indicate the interplay of shoreline change rates and regional-scale climate trends. For example, at clusters 3-2 and 3-3 and clusters 5-2, 5-4 and 5-5 of the erosional coast a slowdown of retreat during the 2006–2014 and 2014–2016 periods correlate with attenuation of HMP during these periods. At the accretional coasts, an increase in the mean rate in 1982–1988 and 1988–2006 coincides with a growth of HMP. Only two clusters of the accretional coast (clusters 5-3 and 5-5) correlate well with HMP change for almost all periods.

Thus, the southwestern part of the erosional stretch (clusters 3-2, 3-3) was partly (at least in 2006–2016) influenced by change in HMP. This is a section of high (up to 9 m) ice-rich permafrost bluffs with relatively narrow (20–30 m) beach and tidal flat, composed mainly of marine loams with high ice content. Such type of coast may be more vulnerable to hydrometeorological forcing, especially to thermal effect, due to its high ice content. At the accretional stretch, a large southwestern part (clusters 5-3 and 5-5) appeared to be subjected to HMP changes. This coast is low-lying (less than 2 m elevation), it differs from more northern coasts by the narrower beach and steeper beach slope. Such laida's coasts may be vulnerable to growth of HMP (mostly WWE) due to its small height, which is common for the laidas of Ural coast of Baydara Bay as well [11] and for the other laidas of the Kara Sea coasts [86].

Moreover, a negligible correlation between mean annual values of HMP and shoreline change rates may be due to the metachronous interplay of wind–wave and thermal impact, leading to lag times of thermal abrasion and thermal denudation [85]. Years with a peak thermal energy may have no geomorphologically significant wave events and thaw sediments would not be eroded. Additionally, previous studies revealed significant changes in intra-annual distribution of wave energy [87]. The peak rates of storm surges may tend to reduce against the background of increasing annual wave forcing due to the sea-ice period extension [87]. In view of the revealed intensification of the thermal action (Figure 13c,d), the absence of thermal-erosional niches during fieldwork, conducted in 2022, may be declared as a geomorphic confirmation of this pattern.

Change in coastal dynamics over time may also depend not only on HMP, but on inner factors of the coastal system. For example, transects of the most widespread first clusters (3-1) of both the erosional and the accretional coasts have quite similar dynamics, but with opposite values: periods of higher rates on the erosional coast correspond to the lower values on the accretional one (Figure 10). That may be related to alongshore sediment fluxes from erosional sections of southwestern coasts to accretional reaches of the northeastern part of the study area.

It is remarkable, that although the mean rate of retreat of the erosional coast tends to reduce, the majority of transects of this coast, forming the most populated cluster 3-1 (as well as both clusters 5-1 and 5-3), show a tendency to a slight acceleration of retreat over time (Figure 10). That probably means that coastal behavior is mainly exerted by change in climatic conditions, mainly by the intensification of the thermal action (Figure 13c,d). Further permafrost thawing [88,89] and wave-energy acceleration [32,40] would rather increase coastal erosion at the regional scale.

### 5.4. Human Impact

It might be suggested that the absence of a clear correlation between coastal dynamics and environmental drivers is due to human interaction within the studied coastal zone. The technogenic impact may influence substantially both spatial and temporal patterns of coastal behavior [90–93]. Such human-triggered erosion of the Kara Sea coasts was observed in previous studies [11,76,94]. Acceleration of retreat rates were triggered by direct (sand excavation from coastal zone, terrain transformation and vegetation disturbance of the hinterland) and/or indirect (disturbance of sediment balance in the coastal zone, disturbance of thermal regime and permafrost sediment features, pollutions) influence. Positive consequences for coastal dynamics in case of appropriate coastal management were also described [95,96]. The shoreline change rates of coastal stretches with no substantial human impact correlate better with hydrometeorological parameters [10,78].

Development of the Kharasavey area started in 1976. In the following years, serious landscape transformation was completed including the excavation of sediment from beach and tideflat for purpose of settlement construction, dredging for shipping, etc. [20,43]. In the period of 1977–1982, a dramatic intensification of coastal retreat at the erosional coast occurred, especially at the southwestern part of the study area (in the alignment of Cape Kharasavey) on permafrost bluffs that composed of the icy loams (cluster 3-3). All accretional coasts (with exception in the alignment of Cape Burunniy) are also experienced drastic retreat during this period that may be due to initiation of the hydraulic sand fill works on the beach and laida surface. Later, in the 1980s, development of the area continued and expanded. At the hinterland, roads inland to the gas field and aerodrome on the beach were constructed. In 1982–1988, the rates of retreat continued to rise and reached their maximum at the erosional stretch, despite the fact that the calculated HMP was relatively low during this period (Figure 11).

In particular, robust erosion at the southwestern part of the coast (clusters 3-2 and 3-3) occurred due to the conducted sediment excavation [20]. Retreat rates were also intensified because of the high susceptibility of the reaches that composed of icy loams. As a result, this study revealed the highest standard deviation of coastal change rates during this period. It may indicate the domination of local-scale drivers (predominantly technogenic) over regional-scale climate change. In the period of 1988–2006, HMP increased, that could trigger activation of retreat of accretional coast (Figure 12). Erosional coast in general experienced attenuation of erosion during this period. That may be relative (against the background of intensification at previous periods), but also may be related to minimal technogenic influence, as in the 1990s development of gas field was halted. In the previous decade (2006–2014, 2014–2016), the rates of retreat/advance were rather stable and approximately equal to the long-term annual average (approximately −1 m/yr at the erosional reaches and −0.5 m/yr at the accretional coasts), as well as HMP and technogenic influence were relatively low (Figure 12).

During the last observed period (that is 2020–2022 for erosional stretch and 2014–2022 for accretional stretch), the rates of erosion slightly increased again which may be connected with the resumption of field development [97]. In the absence of data regarding HMP rates among the last years (2020–2022), it is problematic to determine coastal response to regional-scale drivers. However, after expansion of infrastructure northward to the settlement intensification of coastal retreat rate is observed. Extremely high (up to −6.4 m/yr) erosion found out on the accretional reaches in the alignment of new aerodrome construction (transects 100–160) and to the south from Burunniy Cape (transects 186–206), where sediment excavations from the surface of beach and laida are required for further engineering. Cluster analysis also showed outlier reach (cluster 5-4) of the erosional coast that has experienced rapid regression of retreat rates. This segment of the Kharasavey Cape was an erosional hotspot with peak rates of retreat of up to −6.0 m/yr (Figures 8a and 10c). Shoreline change analysis revealed almost extremely low erosional rates (up to −0.1 m/yr) that may be exerted by beach nourishment during dredging works near the seaport. Ac-

cording to field data, the bluff face of this coastal reach has reduced steepness and is partly vegetated while the beach width reaches peak values among erosional stretch.

The coastal dynamics of the Kharasavey area are greatly affected by human interaction. The coastal change rates of the accretional coasts have non-linear, cyclical behavior (Table 3) that might be due both to transformation of sediment drifts during ice-free period extension and spatiotemporal heterogeneity of technogenic influence. A slight trend to reduction in the erosional rates (Table 2) among erosional stretch that is revealed by shoreline change analysis may be exerted by recession in human impact on the coastal zone. At the same time, cluster analysis revealed a slight trend towards acceleration of coastal retreat among significant parts of the non-modified permafrost bluffs. Extreme shoreline change in human-altered erosional hotspots can lead to the exclusion of the environmental trend at a multi-decadal scale. Thus, shoreline change analysis should be supported by cross-proxy and intra-decadal examination to prevent the loss of subtler, but the most common pattern of shoreline behavior.

## 6. Conclusions

The presented study is the first to quantify spatial and temporal variability of the coastal change rates along both erosional and accretional stretches at Kharasavey key site. Based on remote sensing data, shoreline change analysis was performed and supported by cluster analysis and wind–wave modelling via the Popov–Sovershaev method and WWIII. Cross-proxy examination provides a better understanding of the interplay between coastal behavior and environmental or anthropogenic drivers and might be used for both local shoreline management and further investigation of the Arctic coastline worldwide.

1.  Both ice-rich permafrost bluffs and accretional sandy beaches exhibited a tendency towards persistent erosion. For the studied period (1964–2022), multi-decadal average rate of shoreline change is $-1.03$ m/yr for the erosional stretch and $-0.42$ m/yr for the accretional stretch. The presented rates are comparable to the other key sites with similar geomorphic features in the Kara Sea region.

2.  Shoreline change analysis revealed significant spatial and temporal variability of coastal retreat of the erosional stretch. The southwestern area in the alignment of Cape Kharasavey retains to be an erosional hotspot with peak retreat rates of up to $-7.9$ m/yr (for the period of 1982–1988). A slight recession of the annual average retreat rates was observed in recent decades. Yet, cluster analysis captured a progressive erosion along 71% of the erosional stretch. It might be due to the regional-scale intensification of wind–wave and thermal impact under conditions of ice-free period extension and permafrost thawing. In turn, the peak rates of coastal erosion are determined by local-scale, predominantly technogenic drivers. This study shows that appropriate examination of coastline behavior demands cross-proxy and intra-decadal observation of shoreline movements in order to separate contributions of long-term, regional and temporal, local-scale factors.

3.  On the contrary, non-linear, cyclical behavior of the accretional stretch is exerted by drastic changes in predominant winds and further reversals of the sediment fluxes during ice-free period extension. A total of 94% of the sandy beaches show a tendency to retreat. The peak rates of shoreline collapse were $-8.99$ m/yr (for the period of 1982–1988) and probably caused by initiation of the hydraulic sand fill works. Shoreline progradation (up to several tens of m/yr) was revealed locally, in the alignment of Cape Burunniy. Northeastern migration of this coastal reach may also be related to spit formation under the conditions of wind–wave energy fluctuations responding sea-ice decline.

4.  Comparative analysis of the shoreline rate change and the lithological structure revealed the highest erodibility of the permafrost bluffs that are composed of ice-rich marine loams (up to $-2.75$ m/yr at multi-decadal scale). However, average coastal change is similar among distinct composition ($-1.0$ m/yr for marine loams, $-0.9$ m/yr for marine and lacustrine sands). Assessment of geomorphic features

suggested no clear cause-effect patterns. Correlation between shoreline behavior and coastal geomorphology is limited by low resolution and accuracy of applied data and/or by non-linear interplay of the geomorphic processes (thermal abrasion, thermal denudation, aeolian sedimentation, beach accretion, etc.) that contribute onshore morphology. High-resolution field data are required in order to further assessment of these drivers.

5. Apart from the acceleration of the coastal erosion along the prevailing part of the coastline, no clear coherence of the wave and thermal impact with coastal change rates has been observed. It may be related to both human intervention, metachronous interplay of wind–wave and thermal impact and to the recession of peak wind rates during geomorphologically significant storm surges. This study revealed a rapid increase in the thermal impact in recent decades, while field investigation showed the absence of thermal-erosional niches and predominance of thermal denudation landforms. At the decadal scale, lack of the extreme wind rates is fully compensated by ice-free period extension and acceleration of thermal action. These climate changes coupled with growing exploration and development of the Arctic zone make investigation of the coastal behavior critically important.

**Author Contributions:** Conceptualization, A.N. and G.K.; investigation, G.K. and A.N., methodology, A.N., N.S. and S.M.; software, M.D., P.S. and S.M.; writing—original draft preparation, G.K., A.N. and N.S.; writing—review and editing, G.K. and N.B.; visualization, A.N., G.K. and M.D.; project management, S.O. All authors have read and agreed to the published version of the manuscript.

**Funding:** This research was funded by Russian Science Foundation, grant number 22-17-00097.

**Data Availability Statement:** Data available on request due to privacy restrictions.

**Acknowledgments:** Partially, satellite images are the courtesy of Geoportal of the Moscow State University.

**Conflicts of Interest:** The authors declare no conflict of interest.

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
