# Peer review of "Coastal Dynamics at Kharasavey Key Site, Kara Sea, Based on Remote Sensing Data"

_remotesensing, doi:10.3390/rs15174199_

Round 1

Reviewer 1 Report

The manuscript “Coastal dynamics at Kharasavey key site, Kara Sea, based on remote sensing data” has been reviewed. The work estimates the coastal dynamic using long term remote sensing data, with the help of cluster analysis and wind-wave modelling. The results are interesting as the authors demonstrated the tendency of continuous erosion over the region, with certain exceptions happening due to the particular forcing. The analysis is informatics and detail, such as the permafrost bluffs and accretion sandy beaches providing a good inside understanding. Hence, I suggested the paper be accepted for publication in “Remote Sensing” journal in the present form. 

Author Response

We would like to express thanks for the interest in our work and for approving the article! 

Reviewer 2 Report

This is a very valuable study that examines coastal dynamics over many years and from different perspectives. I congratulate the authors for this comprehensive research. The article is well designed and explained in a way that should be of interest to the readers. However, the following simple corrections would make the paper more understandable.

Line 31: I think that the literature review is missing in the introduction. Past studies should be reviewed and it should be explained which gap in the literature this study will address. Additions to the introduction are necessary.

Line 181: Please add a workflow chart where we can see the process steps of your work in detail.

Line 230: You can write more descriptive about DSAS and the EPR you used. At least we would like to see the formula for calculating the EPR.

Line 381: In which unit is rates of threat/advance expressed in the figure? I can understand that it is m/yr but we would like to see it on the Figure.

Line 384: We would also like to see units in the table.

Lines 389, 391: It would be better to show R2 values as superscript. We can generalize this for the whole manuscript.

Line 781: You can include the limitations of the study in the conclusion section.

Author Response

We wish to express our appreciation for your congarulations and comments! Please follow the attachment file to see a point-by-point response to your corrections.

Reviewer 3 Report

All evaluations have been made in the supplementary file. Please see the supplemental file.

The language of the manuscript needs minor corrections.

Author Response

We wish to express our appreciation for your in-depth corrections and comments! Please follow the attachment file to see a point-by-point response to your suggestions.
